# FAPO: Flawed-Aware Policy Optimization for Efficient and Reliable Reasoning

**Yuyang Ding**
Soochow University
yyding23@stu.suda.edu.cn

**Chi Zhang**
ByteDance Seed
zhangchi.usc1992@bytedance.com

**Juntao Li**[*]
Soochow University
ljt@suda.edu.cn

**Haibin Lin**
ByteDance Seed
haibin.lin@bytedance.com

**Xin Liu**
ByteDance Seed
liuxin.ai@bytedance.com

**Min Zhang**
Soochow University
minzhang@suda.edu.cn

Project Page: https://fapo-rl.github.io

## Abstract

Reinforcement learning with verifiable rewards (RLVR) has emerged as a promising paradigm for enhancing the reasoning capabilities of large language models (LLMs). In this context, models explore reasoning trajectories and exploit rollouts with correct answers as positive signals for policy optimization. However, these rollouts might involve flawed patterns such as answer-guessing and jump-in-reasoning. Such flawed-positive rollouts are rewarded identically to fully correct ones, causing policy models to internalize these unreliable reasoning patterns. In this work, we first conduct a systematic study of flawed-positive rollouts in RL and find that they enable rapid capability gains during the early optimization stage, while constraining reasoning capability later by reinforcing unreliable patterns. Building on these insights, we propose **F**lawed-**A**ware **P**olicy **O**ptimization (**FAPO**), which presents a parameter-free reward penalty for flawed-positive rollouts, enabling the policy to leverage them as useful shortcuts in the warm-up stage, securing stable early gains, while gradually shifting optimization toward reliable reasoning in the later refinement stage. To accurately and comprehensively detect flawed-positive rollouts, we introduce a generative reward model (GenRM) with a process-level reward that precisely localizes reasoning errors. Experiments show that FAPO is effective in broad domains, improving outcome correctness, process reliability, and training stability without increasing the token budget.

## 1 Introduction

Large language models (LLMs) with strong reasoning capabilities, such as OpenAI o-series (OpenAI, 2024; 2025), Deepseek R1 (Guo et al., 2025), have sparked significant attention in reinforcement learning with verifiable rewards (RLVR) (Shao et al., 2024). In this paradigm, models are optimized through rule-based outcome rewards, typically a binary signal indicating whether the final answer is correct, in verifiable tasks like mathematical reasoning (Yu et al., 2025; Team et al., 2025) and code generation (Xiaomi et al., 2025). During RL training, the model explores diverse reasoning trajectories and exploits those with correct final answers as positive signals for policy optimization. This exploration–exploitation paradigm enables LLMs to evolve strong reasoning behaviors, such as planning, which in turn facilitate generalization across a wide range of domains (Huan et al., 2025).

However, certain flawed reasoning patterns could also be reinforced during policy optimization. Recent studies (Zheng et al., 2024; Kalai et al., 2025) have revealed notable flawed reasoning patterns in current LLMs, such as answer-guessing and jump-in-reasoning (Wang et al., 2025), where models reach correct answers through shortcuts. This presents a fundamental challenge for RLVR, i.e., rule-based outcome rewards assign identical positive signals to both flawed-positive and fully correct rollouts, thereby potentially reinforcing unreliable reasoning. This raises an urgent need for (1) *analyzing the distribution and impact of flawed positives throughout the RL process*, and (2) *developing effective mitigation strategies to ensure efficient and reliable reasoning*.

---

[*]Corresponding author

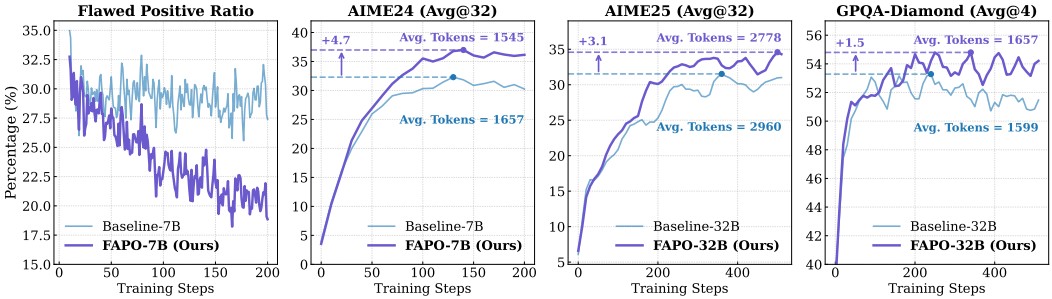

Figure 1: Flawed-positive ratio and performance comparison between FAPO models and baselines.

To this end, we first conduct a preliminary study to investigate the prevalence and impact of flawed positives in the RL process. Our findings indicate that flawed positives persist steadily throughout training: (1) in the early stages, when models are not yet capable of producing fully correct rollouts, flawed positives serve as shortcuts to correct answers, accelerating capability gains; and (2) once the model can generate fully correct rollouts, these flawed positives may hinder learning by reinforcing unreliable reasoning patterns. Thus, the optimal role of flawed positives is to act as stepping stones toward reliable reasoning. Building on these insights, we propose Flawed-Aware Policy Optimization (FAPO), which presents a parameter-free reward penalty to flawed-positive rollouts. FAPO establishes a natural self-exploration learning trajectory: the model initially exploits flawed positives for knowledge, but as its capabilities advance, the training objective gradually shifts toward genuine problem-solving, improving both the training efficiency and reasoning reliability. Furthermore, to accurately and comprehensively detect these flawed positives, we introduce a generative reward model with a process reward, which enables the model to locate intermediate process errors.

Experimental results highlight the strong potential of FAPO. For flawed positive detection, our trained model, FAPO-GenRM-4B, achieves substantial gains on both our newly constructed benchmark, FlawedPositiveBench, and the public ProcessBench (Zheng et al., 2024). When integrated into the final RL process (results in Figure 1), FAPO effectively penalizes flawed-positive rollouts, reducing unreliable reasoning patterns (left subfigure, the decreasing flawed positive ratio), while delivering remarkable improvements across AIME24, AIME25, and GPQA-Diamond (Rein et al., 2024) (other three subfigures), **with clear advantages at nearly all intermediate evaluation checkpoints.** Overall, FAPO offers clear advantages: it enhances outcome correctness, improves process reliability, and training efficiency and stability, all without increasing the token budget.

## 2 PRELIMINARY: UNVEILING FLAWED POSITIVES IN RL TRAINING

### 2.1 PROBLEM DEFINITION AND MOTIVATION

**Group Relative Policy Optimization (GRPO)** GRPO (Shao et al., 2024) is an efficient policy gradient method for LLM reinforcement learning that estimates advantages in a group-relative manner without relying on a learned value model. For a given question $q$, the behavior policy generates a group of $G$ rollouts $\{o_i\}_{i=1}^{G}$, evaluates their rewards $\{R_i\}_{i=1}^{G}$, and normalizes them to obtain per-token advantage estimates $\hat{A}_{i,t}$:

$$\hat{A}_{i,t} = \frac{r_i - \text{mean}(\{R_i\}_{i=1}^{G})}{\text{std}(\{R_i\}_{i=1}^{G})}. \tag{1}$$

The policy model is then updated by maximizing the following clipped surrogate objective:

$$\mathcal{J}_{\text{GRPO}}(\theta) = \mathbb{E}_{(q,a)\sim\mathcal{D}, \{o_i\}_{i=1}^{G}\sim\pi_{\theta_{\text{old}}}(\cdot|q)}$$

$$\frac{1}{G}\sum_{i=1}^{G}\frac{1}{|o_i|}\sum_{t=1}^{|o_i|}\left\{\min\left[\frac{\pi_\theta(o_t|q, o_{<t})}{\pi_{\theta_{\text{old}}}(o_t|q, o_{<t})}\hat{A}_{i,t}, \text{clip}(\frac{\pi_\theta(o_t|q, o_{<t})}{\pi_{\theta_{\text{old}}}(o_t|q, o_{<t})}, 1-\epsilon, 1+\epsilon)\hat{A}_{i,t}\right]\right\}, \tag{2}$$

where $(q, a)$ denotes a question-answer pair sampled from the data distribution $\mathcal{D}$, $\pi_{\theta_{\text{old}}}$ is the old policy, and $\epsilon$ controls the clipping range in importance sampling (Schulman et al., 2017) for stability.

In this work, we adopt several effective strategies such as *clip-higher*, *token-level loss*, and *overlong reward shaping* (Yu et al., 2025), to ensure stable and efficient policy optimization.

$$\mathcal{J}(\theta) = \mathbb{E}_{(q,a)\sim\mathcal{D}, \{o_i\}_{i=1}^G \sim \pi_{\theta_{\text{old}}}(\cdot|q)}$$

$$\frac{1}{\sum_{i=1}^G |o_i|} \sum_{i=1}^G \sum_{t=1}^{|o_i|} \left\{ \min\left[ \frac{\pi_\theta(o_t|q,o_{<t})}{\pi_{\theta_{\text{old}}}(o_t|q,o_{<t})} \hat{A}_{i,t}, \text{clip}\left(\frac{\pi_\theta(o_t|q,o_{<t})}{\pi_{\theta_{\text{old}}}(o_t|q,o_{<t})}, 1-\epsilon_l, 1+\epsilon_h\right) \hat{A}_{i,t} \right] \right\}. \tag{3}$$

In these algorithms, the reward $R$ is the primary supervision signal that guides the policy optimization, and existing RLVR approaches (Yang et al., 2025; Liu et al., 2025b) commonly employ a rule-based outcome reward to mitigate reward hacking (Gao et al., 2023; Weng, 2024), i.e.,

$$R_{\text{RLVR}} = R_{\text{rule}}(o, a^*) = \begin{cases} 1, & \text{If } \mathcal{I}(o, a^*) \\ -1, & \text{Otherwise} \end{cases}, \tag{4}$$

where $\mathcal{I}(o, a^*)$ is an indicator function that returns `True` if the predicted answer extracted from rollout $o$ matches the ground-truth answer $a^*$, and `False` otherwise.

**Flawed Positive Issues** Recent studies (Zheng et al., 2024; Zhang et al., 2025b) have identified notable flawed-positive issues in current LLMs, in some cases even accounting for a ratio of up to 50%, where models may reach correct final answers through unreliable reasoning patterns such as answer-guessing and jump-in-reasoning (Wang et al., 2025). This poses a fundamental challenge for reinforcement learning: rule-based reward functions assign positive signals to flawed-positive rollouts, thereby reinforcing unreliable reasoning patterns and ultimately limiting the model's performance ceiling. Formally, given a question $q$ and reasoning trajectory $\mathbf{x} = [x_1, x_2, \ldots, x_n]$ generated by policy $\pi$, with predicted answer $\hat{a}_\pi$, the rollout is *flawed positive* if

$$\hat{a}_\pi = a^* \text{ and } \exists\, t \in \{1, 2, \ldots, n\} \text{ s.t. step } x_t \text{ is logically invalid.} \tag{5}$$

While prior works have primarily revealed the prevalence of these issues in benchmark evaluations, their underlying mechanisms and impact on the RL process remain largely underexplored.

## 2.2 FLAWED POSITIVE ANALYSIS IN REINFORCEMENT LEARNING

**Flawed Positives are Prevalent in Initial Checkpoints** We first examine flawed positive issues in current LLMs, as they establish the starting conditions for subsequent RL optimization. We evaluate three representative models: Qwen2.5-Math-7B-Base (Yang et al., 2024b), Llama3.3-70B-Instruct (Dubey et al., 2024), and Qwen3-1.7B (Yang et al., 2025), on the DAPO-Math dataset. Following Zhang et al. (2025b), we employ Qwen3-32B to determine whether the reasoning trajectory contains unreliable reasoning patterns. As shown in Figure 2 (a), flawed positives are prevalent across various LLMs, accounting for 20%–40% of correct rollouts, highlighting the severity of this issue. Beyond the automatic LLM-as-a-judge evaluation, we also conduct a manual case study of flawed-positive samples and analyze their underlying causes, which are provided in Appendix G.

**Flawed Positives are Stepping Stones in Learning** Reinforcement learning is often formulated as an end-to-end optimization process driven by self-exploration towards self-improvement. To better understand this process, and in particular the role of flawed-positive rollouts, we design a simulated experiment in which the learning stage of each sample is approximated by its rollout accuracy. Specifically, we use a model to generate multiple rollouts per sample, compute the corresponding rollout accuracy, and then group all samples into different learning stages, as illustrated in Figure 2 (b). The results reveal a clear trend: flawed positives are most prevalent during the early learning stages but diminish significantly as training progresses. This highlights their expected role as natural stepping stones in the learning trajectory, allowing the model to initially reach correct answers before gradually evolving the capability to produce fully correct solutions.

**Flawed Positives Persist and Exert Twofold Effects** We further train a pre-trained model, Qwen2.5-Math-7B, with RL on DAPO-Math, and track its learning trajectory, as shown in Figure 2 (c). While the model's rollout accuracy steadily improves, the flawed-positive ratio remains almost constant at around 30%. This indicates that the optimization process struggles to shift from unreliable reasoning to genuine problem-solving. A major concern is that flawed positives receive the

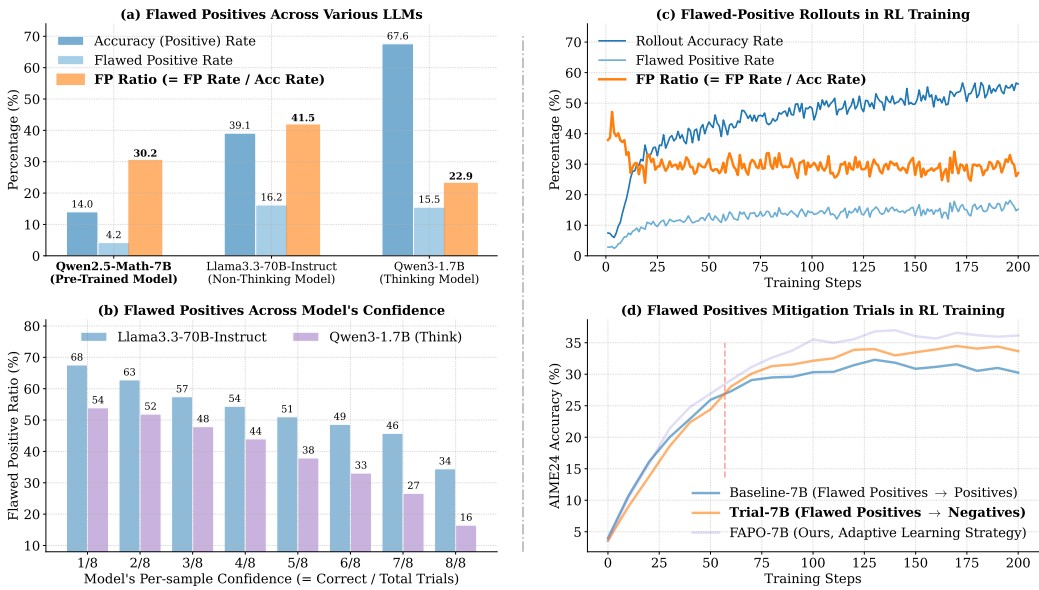

Figure 2: Preliminary experiment results of flawed positives.

same rewards as correct solutions, thereby reinforcing unreliable reasoning patterns and hindering progress. To further explore this, we conduct preliminary trials using Qwen3-32B to detect flawed positives and assign them negative signals (same as negative rollouts) during training. Figure 2 (d) reports performance on AIME24 throughout the training process. Compared with the baseline RLVR setting (blue line), penalizing flawed positives (orange line) yields significant performance gains, though improvements emerge more gradually in the early stages. From the above findings, we find flawed positives persist throughout training, and exert a twofold effect: **(1) flawed positives act as stepping stones, enabling the model to achieve rapid capability gains in the early stages, and (2) their improper reward assignment can trap optimization in unreliable reasoning.**

## 3    FAPO: FLAWED-AWARE POLICY OPTIMIZATION

Building on these insights, we propose **F**lawed-**A**ware **P**olicy **O**ptimization (FAPO) algorithm. For flawed-positive detection, directly employing a strong LLM like Qwen3-32B is impractical and computationally inefficient. Instead, we propose an RL algorithm to train a compact yet effective generative reward model (GenRM). We then present an adaptive learning algorithm that dynamically adjusts advantage assignment towards the current suitable optimization direction.

### 3.1    FLAWED POSITIVE DETECTION

**Evaluating Flawed Positive Detection Capabilities of LLMs**    To identify a suitable LLM that can detect flawed positives both effectively and efficiently, we construct an evaluation set, *Flawed-PositiveBench*, by collecting positive samples (including flawed ones) from ProcessBench (Zheng et al., 2024). We then quantify the detection capability with the following three metrics:

$$precision = \frac{\#\{\hat{y}_\theta = y^* = \text{FP}\}}{\#\{\hat{y}_\theta = \text{FP}\}}, \; recall = \frac{\#\{\hat{y}_\theta = y^* = \text{FP}\}}{\#\{y^* = \text{FP}\}}, F_1 = \frac{2}{1/precision + 1/recall}, \; (6)$$

where $\hat{y}_\theta$ indicates whether the judge model $\theta$ predicts a response as a flawed positive (FP), and $y^*$ is the ground-truth label. Precision reflects the correctness of FP predictions, recall measures the coverage of true FPs, and the $F_1$ score provides a balanced summary of both. As shown in Figure 3, we observe that many models, such as Qwen3-4B-Instruct (Yang et al., 2025) and Qwen2.5-Math-PRM-72B (Zhang et al., 2025b), exhibit an over-critic phenomenon: they achieve high recall but suffer from low precision. Closer inspection reveals that these models often overemphasize minor or unnecessary errors like unsimplified fractions. Overall, lightweight models struggle to provide

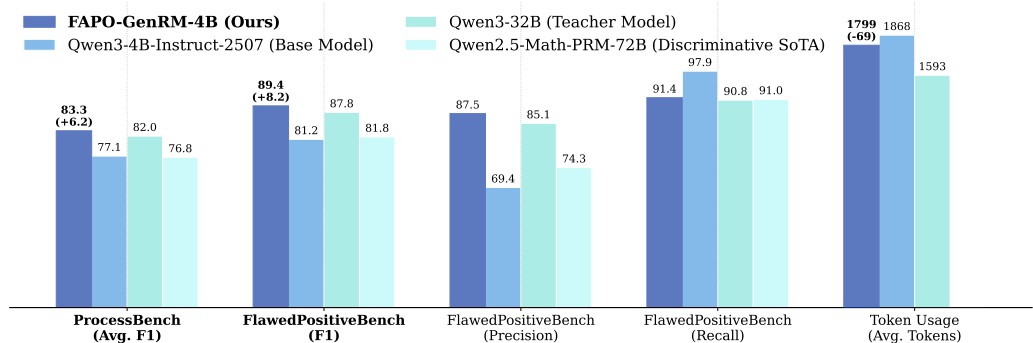

Figure 3: Performance of current state-of-the-art (SoTA) generative models and discriminative PRMs. Detailed subset-level results and additional models are reported in Table 2.

appropriate criticisms to detect flawed positives, while stronger models achieve better accuracy but remain impractical for online RL use due to slow inference. **These findings suggest that existing models are not well-aligned in both detecting capabilities and inference efficiency.**

**Enhancing Detection Capabilities via Step-wise RL Optimization** To enhance the detection capability, we introduce a step-wise RL reward formulation. Starting from a generative model, we develop the following RL strategies for training a generative reward model (GenRM):

$$R_{\text{FAPO-GenRM}} = R_{\text{Outcome}} + \mathbf{R_{\text{Process}}}$$

$$\text{where } R_{\text{Outcome}} = \begin{cases} 1, & \text{If } \hat{y}_\theta = y^* \\ -1, & \text{Otherwise} \end{cases}, \quad R_{\text{Process}} = \begin{cases} -\frac{|\hat{t}_\theta - t^*|}{n}, & \text{If } \hat{y}_\theta = y^* = \text{FP} \\ 0, & \text{Otherwise} \end{cases}. \quad (7)$$

Building upon the outcome reward, we introduce a step-wise penalty, $R_{\text{Process}}$, for fine-grained and step-wise optimization. Here, $\hat{t}_\theta$ and $t^*$ denote the predicted and ground-truth error indices, and $n$ is the total number of steps, ensuring $R_{\text{Process}} \in [-1, 0]$. In flawed-positive cases, the penalty is distance-sensitive: predictions closer to the true error receive higher rewards, while those farther away incur stronger penalties. This design guides the model toward precise error localization and fosters genuine error-detection ability, rather than mere guessing, based on the two key points:

- **Learning beyond guessing:** Flawed-positive rollouts also exist in the error detection task, particularly in the yes/no setting, where the model can often guess the label without truly identifying errors. Such guessing rollouts offer little optimization benefit. To mitigate this, we introduce the penalized step reward that guides the model toward genuine critic capabilities rather than guessing.
- **Natural reward shift:** In early training, the reward design naturally emphasizes prediction correctness, as improving $R_{\text{Baseline}}$ yields substantial gains ($-1 \rightarrow 1$), whereas $R_{\text{Process}}$ provides only limited gains ($R_{\text{Process}} \in [-1, 0]$). As correctness saturates, process optimization becomes increasingly prioritized. This enables a smooth transition without additional controlling hyperparameters.

## 3.2 Flawed Positive Penalization

With the GenRM detecting flawed positives, we then regulate their roles in the final RL optimization. As discussed in Section 2.2, flawed positives should ideally facilitate rapid warm-up and then be penalized to enable genuine problem-solving. However, the key challenge lies in *how to balance their encouragement and suppression and when to shift between these roles*. To address this, we introduce a reward-penalization mechanism with a group-relative advantage estimation:

$$R_{\text{FAPO}}(o, a^* | \theta) = R_{\text{RLVR}}(o, a^*) + \mathbf{R_\Delta(\mathbf{o}, \mathbf{a}^* | \theta)},$$

$$\text{where } R_\Delta(o, a^* | \theta) = \begin{cases} -\lambda, & \text{If } \mathcal{I}(o, a^*) \text{ and } \hat{y}_\theta(o, a^*) = \text{FP} \\ 0, & \text{Otherwise} \end{cases}, \quad (8)$$

$$\hat{A}_{i,t} = \left[ r_i - \text{mean}(\{R_i\}_{i=1}^G) \right] / \text{std}(\{R_i\}_{i=1}^G).$$

where $R_{\text{RLVR}}$ denotes the standard baseline (defined in Equation 4), and $\lambda$ controls the penalization strength. To better characterize the entire learning dynamics of FAPO, we provide a theoretical

analysis in Appendix A, which demonstrates how FAPO enables the natural optimization shift while further stabilizing the RL training process. Concretely, when the current rollout stage contains $\alpha$ proportion of positive samples and $\beta$ proportion of negative samples, the optimization shifts from the warm-up stage to the refinement stage once the learning progress $\rho = \frac{\alpha}{\beta}$ reaches $\frac{2}{\lambda} - 1$. Moreover, as optimization continues, when $\rho > \frac{4}{\lambda} - 1$, the estimated advantage for positive samples becomes downscaled, making the optimization more stable. In the process, the value of $\lambda$ determines the timing of this optimization shift. We adopt a majority-guided strategy, where the optimization direction is determined by whether positive or negative samples dominate. This majority-guided strategy yields $\rho_{\text{shift}} = 1$, further deteming $\lambda = 1$. We set $\lambda = 1$ as the default setting. Overall, FAPO provides a principled mechanism for guiding the optimization process, aligning with the ideal learning trajectory where the focus initially lies in producing correct solutions when model capability is limited, and naturally shifts toward refining reliability once correct rollouts surpass incorrect ones.

## 4 EXPERIMENTS

### 4.1 TRAINING DETAILS

In this work, we validate the effectiveness of FAPO on Qwen2.5-Math-7B (Yang et al., 2024b) and Qwen2.5-32B (Yang et al., 2024a). We adopt GRPO (Shao et al., 2024) with several commonly used strategies, including *clip-higher*, *token-level loss*, and *overlong reward shaping* (Yu et al., 2025), as our baseline algorithm. Notably, FAPO can be easily transferred to any other RLVR method as a drop-in replacement for rule-based outcome rewards. We conduct RL training using verl framework (Sheng et al., 2025), and develop an asynchronous architecture that decouples rollout inference and generative reward modeling, which substantially improves training efficiency.

**FAPO-GenRM**  To train the GenRM model via reinforcement learning, we construct a flawed-positive dataset, FAPO-Critic-85K. To ensure broad coverage, we employ a series of models from the LLaMA and Qwen families, ranging from 7B to 70B, to generate multiple responses to questions drawn from DAPO-Math-17K (Yu et al., 2025). Based on these responses, we select the samples with correct final answers and then employ Qwen3-32B to identify the inherent step-level error location. This yields the final process-error dataset: $\mathcal{D}_{\text{FAPO-Critic}} = \{(q_i, r_i, t_i)\}_{i=1}^{N}$, where $t_i$ denotes the first error index of response $r_i$, and and fully correct responses are included with $t_i = +\infty$ for convenience. This dataset is then used to train Qwen3-4B-Instruct (Yang et al., 2025) with the reward defined in Equation 7, and additional hyperparameter settings are provided in Appendix B.

**FAPO-Reasoning**  The trained critic model, FAPO-GenRM-4B, is then used to detect and penalize flawed positives in reinforcement learning for reasoning tasks, optimized with the reward defined in Equation 8. In practice, we deploy the GenRM as an external LLM service on a computing cluster, where process rewards are obtained via remote API requests during RL training. To ensure efficiency, we launch multiple server workers and employ a router to distribute requests with balanced load across workers. This decoupled design enables asynchronous interaction between GenRM and other RL components, substantially improving training efficiency and making the integration of GenRM into large-scale RL training practically feasible. Further details on infrastructure design and hyperparameter configurations are provided in Section 4.5 and Appendix B, respectively.

### 4.2 EVALUATION SETUP

**Flawed-Positive Detection**  We primarily evaluate GenRM on FlawedPositiveBench, whose construction procedure and evaluation metrics are detailed in Section 3.1, as this benchmark directly aligns with our research purpose. In addition, we also include ProcessBench (Zheng et al., 2024), which focuses on recognizing fully correct samples and precisely locating errors in incorrect responses. We report the harmonic mean of the accuracies on correct and erroneous samples. Furthermore, we incorporate several state-of-the-art (SoTA) discriminative and generative models as strong baselines for comprehensive comparison.

**Reasoning Evaluation**  We conduct a comprehensive evaluation covering AIME24 (Math), AIME25 (Math), and GPQA-Diamond (General Domain) (Rein et al., 2024). Rather than limiting the analysis to a single selected checkpoint, we present all intermediate evaluation outcomes

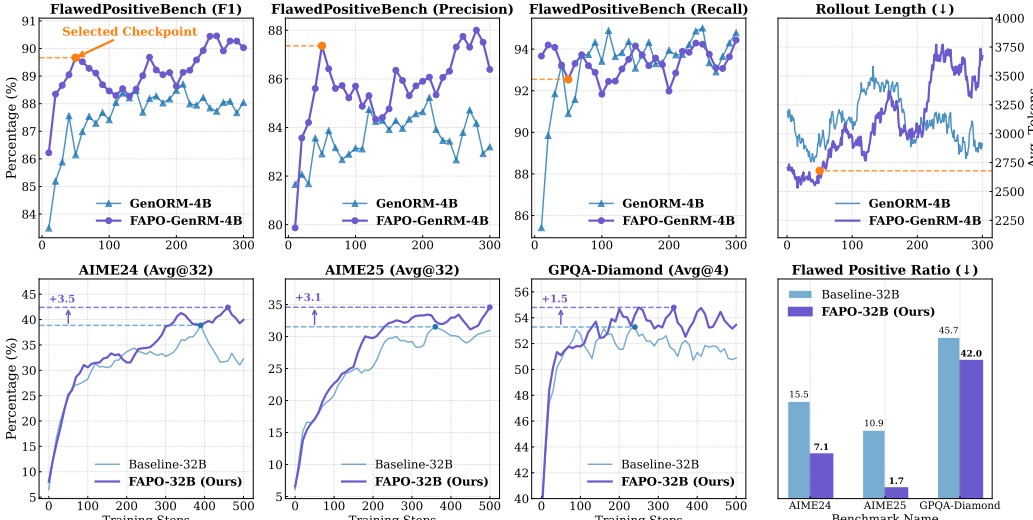

Figure 4: Performance of FAPO-GenRM and FAPO-Reasoning during training. **Top row:** comparison between FAPO-GenRM and the baseline outcome reward models (setup in Equation 7). **Bottom row:** comparison between FAPO-Reasoning and the baseline setting (setup in Equation 8). Detailed results in a broader domain can be seen in Table 2 and Table 3.

throughout the RL process. This not only illustrates the performance gains achieved during the training process but also highlights the stability and scalability of the optimization procedure, thereby providing stronger evidence of the effectiveness and robustness of our approach.

## 4.3 MAIN RESULTS

**FAPO-GenRM Performance** The top row of Figure 4 illustrates the training dynamics of FAPO-GenRM. The model exhibits significant performance gains in the early stages and continues to improve as training progresses. For subsequent use, however, we select an early checkpoint, as it delivers strong results with shorter responses, which is crucial for maintaining efficiency when integrating GenRM into RL training. Figure 3 compares our trained model against state-of-the-art discriminative and generative baselines. Built upon Qwen3-4B-Instruct, our approach achieves substantial improvements on both FlawedPositiveBench and ProcessBench, even outperforming the teacher model Qwen3-32B, further demonstrating the effectiveness of our approach. Additional results of our model and other strong baselines can be checked in Table 2.

**FAPO-Reasoning Performance** Figure 1 and the bottom row of Figure 4 summarize the overall performance of the FAPO reasoning models, which can be highlighted in the following aspects:

- **Outcome Correctness:** Across benchmarks, FAPO consistently maintains a clear advantage over the baselines in both mathematical and general-domain tasks, demonstrating that detecting and penalizing flawed positives leads to broad improvements in problem-solving ability.
- **Process Reliability:** We also measure the proportion of flawed positives. The results show that FAPO responses exhibit a substantially lower flawed-positive ratio. Beyond the LLM-as-a-judge approach using Qwen3-32B, we also launch a manual verification of unreliable reasoning patterns, with details and results in Table 6, demonstrating the effectiveness of FAPO.
- **Training Stability:** By mitigating the impact of flawed positives, training stability is significantly enhanced. The overall learning curves are smoother, and unlike the baselines, FAPO does not exhibit a notable performance drop in the later stages of training.
- **Token Budget:** The improvements from FAPO do not require longer responses. While prior work (Luo et al., 2025; An et al., 2025) has shown that scaling up response length can yield substantial gains, FAPO achieves improvements without relying on this factor.

## 4.4 ABLATION STUDY

**Effectiveness of FAPO-GenRM method**  Previous results have already demonstrated that our GenRM design achieves state-of-the-art (SoTA) performance on the error detection task (see Figure 3). Here, we further evaluate its impact on the final RL process. Figure 5 compares FAPO-GenRM with the base model (Qwen3-4B-Instruct) during RL training. The results indicate that stronger detection capability ultimately translates into improved performance. This highlights two key points: (1) the proposed FlawedPositiveBench provides a reliable measure of detection ability that is well aligned with final performance, and (2) flawed positive detection plays a crucial role, where even small improvements can yield substantial performance gains.

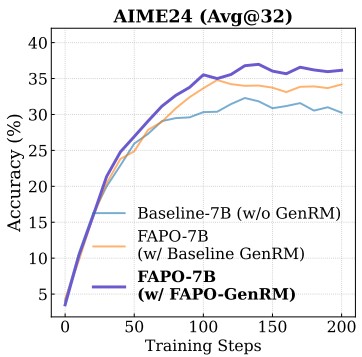

Figure 5: GenRM effectiveness.

**Impact of Self-Correction Capability**  Self-correction is an important evolved mechanism in RL, allowing reasoning models to recover from initial mistakes and eventually reach correct answers. Nevertheless, when correctness depends heavily on long rollouts, it could also be regarded as a form of flawed positives. While self-correction facilitates progress in the early stages of learning, it becomes less desirable once the model can solve problems directly, where fully correct rollouts not only ensure reliability but also promote more efficient reasoning. To illustrate this phenomenon, Figure 6 visualizes rollout length during training: both FAPO and the baseline initially depend on self-correction, but over time, FAPO shifts toward fully correct rollouts, resulting in shorter rollouts, more efficient reasoning, and consistent performance gains. These results indicate that FAPO preserves the benefit of learning from self-corrected rollouts at the beginning, but gradually shifts toward prioritizing fully correct rollouts in later stages.

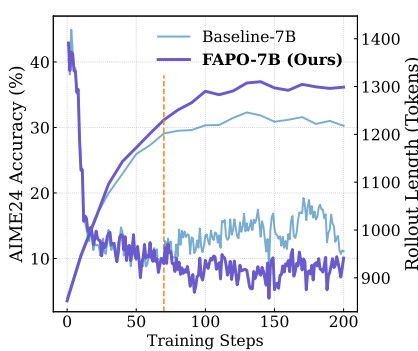

Figure 6: Self-correction Analysis.

### 4.5 DISCUSSION: GENRM APPLICATION IN FUTURE RL SYSTEMS

Introducing generative reward models (GenRMs) may have a considerable impact on the whole RL process, influencing both algorithmic effectiveness and infrastructure efficiency. In this section, we discuss the application potential of GenRMs (with FAPO as an example) in future RL systems, considering perspectives from both algorithmic development and infrastructure design.

**Algorithmic Challenge: Reward Hacking**  The primary algorithmic challenge of deploying GenRM in large-scale RL systems lies in *reward hacking*, where the policy exploits imperfections in the reward signal to achieve high scores without genuinely performing the intended reasoning task. This issue is especially pronounced with complex, fine-grained reward signals produced by reward models, as they provide more opportunities for the policy to discover shortcuts for maximizing rewards. For example, we experiment with a process-based reward that assigns scores according to the proportion of correct steps before the first detected error. However, this design results in a form of reward hacking: the model tends to output only those reasoning steps in which it has very high confidence, while skipping uncertain ones altogether.

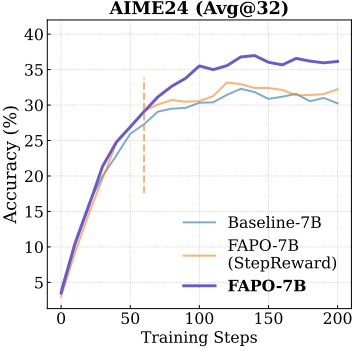

Figure 7: Step reward ablation.

As shown in Figure 7, although the step-ratio reward delivers some capability gains at the early stage compared to the baseline, the subsequent progress stalls due to reward hacking. This behavior causes an obvious *jump-in-reasoning* phenomenon, which is undesirable as it undermines the reliability of the reasoning process.

**Infrastructure Challenge: Long-tail Problem**
The long-tail problem has long been a key bottle-neck in scaling large-scale RL systems (Liang et al., 2018), where GPUs often remain idle during the generation of long-tail samples. A concern with GenRM is that it introduces an additional generation stage, further aggravating this inefficiency. To make FAPO practical for large-scale RL systems, we adopt the following efforts: (1) **Asynchronous design:** We decouple GenRM from rollout inference and actor training, reducing GPU idle time. While a fully synchronous design may offer a better system-level solution (Fu et al., 2025), this lies beyond the primary research focus of our work, and we leave this as an important future work. (2) **GenRM training:** FAPO employs an overlong reward strategy in training and treats token budget as a key criterion in final checkpoint selection. With these efforts, the training time of FAPO is increased by less than 20% relative to the baseline.

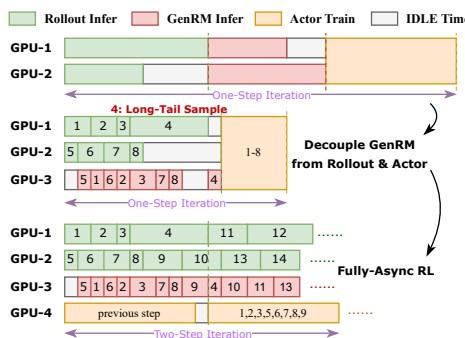

Figure 8: GenRM infrastructure design.

## 5 RELATED WORK

**LLM Reinforcement Learning** Reinforcement learning has emerged as a promising paradigm for advancing LLM reasoning capabilities (Xu et al., 2025a; Zhang et al., 2025a), with the long-term vision of Artificial Superintelligence. Recent milestones (Guo et al., 2025; OpenAI, 2025) demonstrate that RL in verifiable tasks (Liu et al., 2025c; He et al., 2025; Feng et al., 2025) enables models to develop reasoning patterns such as planning and tool usage, which in turn foster generalization across broader domains (Huan et al., 2025; Seed et al., 2025). While verifiable rewards provide only binary feedback, learning progresses gradually. Building on this direction, we explore the role of flawed positives in this process and introduce the Flawed-Aware Policy Optimization (FAPO) algorithm, which promotes a more natural learning trajectory and enables efficient and reliable RL.

**Reward Models in Reinforcement Learning** Reward models offer a promising approach to addressing the limitations of rule-based rewards, which can be broadly categorized into generative (GenRMs) and discriminative reward models (DisRMs). Current GenRMs mainly serve as flexible verifiers (Xu et al., 2025b; Chen et al., 2025) that augment rule-based systems when correctness cannot be reliably assessed by predefined rules (Liu et al., 2025a; Zhao et al., 2025). In addition to verifiable reasoning tasks, GenRMs are also applied in non-verifiable tasks, providing subjective and rubric-based rewards (Mahan et al., 2024; Zhang et al., 2024; Zhou et al., 2025). In contrast, DisRMs typically output fine-grained, dense rewards at every decision step, such as token-level (Rafailov et al., 2023; Cui et al., 2025), step-level (Lightman et al., 2023; Wang et al., 2023; Ding et al., 2025), and turn-level (Qian et al., 2025; Dong et al., 2025), to support more precise optimization. However, the complexity of such dense rewards also makes them vulnerable to reward hacking (Gao et al., 2023), as policies may exploit spurious shortcuts instead of learning the intended behaviors (Weng, 2024), limiting their application in large RL systems. To address this challenge, FAPO introduces an interpretable framework that trains GenRMs to detect flawed positives and provide nuanced, interpretable rewards. Both empirical experiments (Appendix 4) and theoretical analysis (Section A) demonstrate that FAPO exhibits strong robustness against reward hacking and scales effectively.

## 6 CONCLUSION

This paper introduces the Flawed-Aware Policy Optimization (FAPO) algorithm to enhance efficiency and reliability in LLM RL. We first reveal the role of flawed-positive rollouts: they accelerate capability gains in the early stage but ultimately constrain reasoning quality by reinforcing unreliable patterns. To reconcile this trade-off, FAPO applies a parameter-free reward adjustment that leverages flawed positives as shortcuts during warm-up while progressively steering optimization toward reliable reasoning. In addition, we develop a generative reward model (GenRM) with process-level signals to accurately detect and localize reasoning errors. Both empirical experiments and theoretical analysis demonstrate the effectiveness of FAPO in future large-scale RL systems.

## ETHICS STATEMENT

We confirm that this work adheres to ethical research practices. All data and LLMs used are publicly available (including API access) and properly cited, with no involvement of human subjects. The Use of LLM statement is illustrated in Appendix E.

## REPRODUCIBILITY STATEMENT

We have made every effort to ensure the reproducibility of the results reported in this paper. Details of the algorithm design are provided in Section 3, while the infrastructure setup is described in Section 4. Additional information, including the dataset, training hyperparameters, and evaluation setups, is presented in Section 4 and Appendix B.

## ACKNOWLEDGEMENT

We want to thank all the anonymous reviewers for their valuable comments. This work was supported by the National Science Foundation of China (NSFC No. 62576232) and the Young Elite Scientists Sponsorship Program by CAST (2023QNRC001).

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

## A  THEORETICAL ANALYSIS OF FAPO ALGORITHM

To better understand the effectiveness of FAPO and the whole learning process, we present a theoretical analysis. We begin by comparing FAPO with baseline settings to illustrate how the optimization direction shifts during the RL process and how the corresponding advantage estimation evolves. In this context, $\lambda$ plays a crucial role in controlling the optimization dynamics. Building on this, we further introduce a parameter-free, majority-guided optimization strategy.

In the context of LLM RL, we typically maximize the following clipped surrogate objective:

$$
\begin{aligned}
\mathcal{J}_{\text{GRPO}}(\theta) = \mathbb{E}_{(q,a)\sim\mathcal{D},\{o_i\}_{i=1}^{G}\sim\pi_{\theta_{\text{old}}}(\cdot|q)} \\
\frac{1}{G}\sum_{i=1}^{G}\frac{1}{|o_i|}\sum_{t=1}^{|o_i|}\left\{\min\left[\frac{\pi_\theta(o_t|q,o_{<t})}{\pi_{\theta_{\text{old}}}(o_t|q,o_{<t})}\hat{A}_{i,t}, \text{clip}(\frac{\pi_\theta(o_t|q,o_{<t})}{\pi_{\theta_{\text{old}}}(o_t|q,o_{<t})},1-\epsilon,1+\epsilon)\hat{A}_{i,t}\right]\right\},
\end{aligned} \tag{9}
$$

with advantage estimated in a group-relative manner:

$$
R_i = \begin{cases} 1, & \text{If } o_i \text{ is correct} \\ -1, & \text{Otherwise} \end{cases}, \quad \hat{A}_{i,t} = \frac{r_i - \text{mean}(\{R_i\}_{i=1}^{G})}{\text{std}(\{R_i\}_{i=1}^{G})}, \tag{10}
$$

In this context, $\mu_{\text{FAPO}} = \text{mean}(\{R_i\}_{i=1}^{G})$ determines the sign of the advantage $\hat{A}_{i,t}$, which in turn dictates the current optimization direction while $\text{std}(\{R_i\}_{i=1}^{G})$ can be regarded as a scaling factor. FAPO introduces the following modification to the reward function:

$$
R_{\text{GRPO}} = \begin{cases} 1, & \text{If } o \text{ is correct} \\ -1, & \text{Otherwise} \end{cases} \xrightarrow{\text{penalty}} R_{\text{FAPO}} = \begin{cases} 1, & \text{If } o \text{ is fully correct} \\ 1-\lambda & \text{If } o \text{ is correct yet flawed} \\ -1, & \text{Otherwise} \end{cases}. \tag{11}
$$

We assume that the current sample contains $n$ rollouts, with a proportion of $\alpha$ fully correct positives and a proportion of $\beta$ negatives, leaving $1 - \alpha - \beta$ as flawed positives. Since GRPO does not distinguish between fully correct and flawed positives, the resulting advantage estimation is:

$$
\begin{aligned}
\mu_{\text{GRPO}} &= \text{mean}(\{R_i\}_{i=1}^{n}) = \frac{1 \times (1-\beta)n + (-1) \times \beta n}{n} = 1 - \beta - \beta = 1 - 2\beta, \\
\sigma_{\text{GRPO}}^2 &= \text{std}(\{R_i\}_{i=1}^{n})^2 = \frac{(1-\mu_{\text{GRPO}})^2 \times (1-\beta)n + (-1-\mu_{\text{GRPO}})^2 \times \beta n}{n} \\
&= (1-\beta)(1-\mu_{\text{GRPO}})^2 + \beta(1+\mu_{\text{GRPO}})^2,
\end{aligned} \tag{12}
$$

while the advantage estimation of FAPO is

$$
\begin{aligned}
\mu_{\text{FAPO}} &= \text{mean}(\{R_i\}_{i=1}^{n}) = \frac{1 \times (\alpha n) + (1-\lambda) \times (n - \alpha n - \beta n) + (-1) \times \beta n}{n} \\
&= \alpha + (1-\lambda)(1-\alpha-\beta) - \beta \\
&= \alpha - \alpha(1-\lambda) - \beta - \beta(1-\lambda) + 1 - \lambda \\
&= 1 - 2\beta - (1-\alpha-\beta)\lambda \\
&= \mu_{\text{GRPO}} - (1-\alpha-\beta)\lambda, \\
\sigma_{\text{FAPO}}^2 &= \text{std}(\{R_i\}_{i=1}^{n})^2 = \frac{(1-\mu_{\text{FAPO}})^2 \times \alpha n + (1-\lambda-\mu_{\text{FAPO}})^2 \times (n-\alpha n-\beta n) + (-1-\mu_{\text{FAPO}})^2 \times \beta n}{n} \\
&= \alpha(1-\mu_{\text{FAPO}})^2 + (1-\alpha-\beta)(1-\lambda-\mu_{\text{FAPO}})^2 + \beta(1+\mu_{\text{FAPO}})^2,
\end{aligned} \tag{13}
$$

For $\sigma_{\text{FAPO}}^2$, we let $\gamma = 1 - \alpha - \beta$, so $\sigma_{\text{FAPO}}^2 = \sigma_{\text{GRPO}}^2 - \gamma\lambda$, then

$$
\begin{aligned}
\sigma_{\text{FAPO}}^2 &= \alpha(1-\mu_{\text{FAPO}})^2 + (1-\alpha-\beta)(1-\lambda-\mu_{\text{FAPO}})^2 + \beta(1+\mu_{\text{FAPO}})^2 \\
&= \alpha(1-\mu_{\text{GRPO}}+\gamma\lambda)^2 + \gamma(1-\mu_{\text{GRPO}}+\gamma\lambda-\lambda)^2 + \beta(1+\mu_{\text{GRPO}}-\gamma\lambda)^2 \\
&= [\alpha(1-\mu_{\text{GRPO}})^2 + \gamma(1-\mu_{\text{GRPO}})^2 + \beta(1+\mu_{\text{GRPO}})^2] \\
&\quad + [2\alpha\gamma\lambda(1-\mu_{\text{GRPO}}) + 2\gamma(1-\mu_{\text{GRPO}})(\gamma\lambda-\lambda) - 2\beta(1+\mu_{\text{GRPO}})\gamma\lambda] \\
&\quad + [\alpha\gamma^2\lambda^2 + \gamma(\gamma-1)^2\lambda^2 + \beta\gamma^2\lambda^2] \\
&= A + B + C
\end{aligned} \tag{14}
$$

We break $\sigma^2_{\text{FAPO}}$ down into three sub-expressions (A, B, C) and simplify them separately.

$$
\begin{aligned}
A &= \alpha(1 - \mu_{\text{GRPO}})^2 + \gamma(1 - \mu_{\text{GRPO}})^2 + \beta(1 + \mu_{\text{GRPO}})^2 \\
&= (1 - \beta)(1 - \mu_{\text{GRPO}})^2 + \beta(1 + \mu_{\text{GRPO}})^2 \quad \rightarrow \because \gamma = 1 - \alpha - \beta \\
&= \sigma^2_{\text{GRPO}} \\
B &= 2\alpha\gamma\lambda(1 - \mu_{\text{GRPO}}) + 2\gamma(1 - \mu_{\text{GRPO}})(\gamma\lambda - \lambda) - 2\beta(1 + \mu_{\text{GRPO}})\gamma\lambda \\
&= 2\gamma\lambda[\alpha(1 - \mu_{\text{GRPO}}) - (1 - \mu_{\text{GRPO}})(1 - \gamma) - \beta(1 + \mu_{\text{GRPO}})] \\
&= 2\gamma\lambda[2\alpha\beta - 2\beta(\alpha + \beta) - \beta(2 - 2\beta)] \quad \rightarrow \because \gamma = 1 - \alpha - \beta, \mu_{\text{GRPO}} = 1 - 2\beta \\
&= 2\gamma\lambda \cdot (-2\beta) = -4\gamma\lambda\beta \\
C &= \alpha\gamma^2\lambda^2 + \gamma(\gamma - 1)^2\lambda^2 + \beta\gamma^2\lambda^2 \\
&= (1 - \gamma)\gamma^2\lambda^2 + \gamma(1 - \gamma)^2\lambda^2 \quad \rightarrow \because \gamma = 1 - \alpha - \beta \\
&= \gamma(1 - \gamma)\lambda^2 \\
B + C &= -4\gamma\lambda\beta + \gamma(1 - \gamma)\lambda^2 = \gamma\lambda(\lambda(\alpha + \beta) - 4\beta) \quad \rightarrow \because \lambda = 1 - \alpha - \beta \\
&= \lambda\gamma(1 - \gamma)(\lambda - \frac{4\beta}{\alpha + \beta}) = \lambda\gamma(1 - \gamma)(\lambda - \frac{4}{\alpha/\beta + 1})
\end{aligned}
\tag{15}
$$

Thus, $\mu_{\text{FAPO}}$ and $\sigma^2_{\text{FAPO}}$ can be expressed in terms of $\mu_{\text{GRPO}}$ and $\sigma^2_{\text{GRPO}}$:

$$
\begin{cases}
\mu_{\text{FAPO}} = \mu_{\text{GRPO}} - \lambda\gamma \\
\sigma^2_{\text{FAPO}} = \sigma^2_{\text{GRPO}} + \lambda\gamma(1 - \gamma)(\lambda - \frac{4}{\alpha/\beta + 1})
\end{cases}
\tag{16}
$$

**When Optimization Direction Shift**   We assume a complete learning process that begins with the model unable to solve any problems, i.e., $\beta = 1$. As training progresses, $\beta$ gradually decreases while $\alpha$ increases. The shift in optimization direction occurs when:

$$
\begin{aligned}
\hat{A}_{\text{Flawed}} &= \frac{1 - \lambda - \mu_{\text{FAPO}}}{\sigma_{\text{FAPO}}} < 0 \\
&\Leftrightarrow \mu_{\text{FAPO}} > 1 - \lambda \\
&\Leftrightarrow \mu_{\text{FAPO}} = \alpha\lambda - \beta(2 - \lambda) + 1 - \lambda > 1 - \lambda \\
&\Leftrightarrow \lambda > \frac{2\beta}{\alpha + \beta} = \frac{2}{\alpha/\beta + 1} \\
&\Leftrightarrow \frac{\alpha}{\beta} > \frac{2}{\lambda} - 1
\end{aligned}
\tag{17}
$$

**How Scaling Factor Changes**   The scaling factor $\sigma^2_{\text{FAPO}}$ changes over $\sigma^2_{\text{GRPO}}$ is:

$$
\begin{aligned}
&\sigma^2_{\text{FAPO}} - \sigma^2_{\text{GRPO}} = \lambda\gamma(1 - \gamma)(\lambda - \frac{4}{\alpha/\beta + 1}) \\
&\text{when } \frac{\alpha}{\beta} < \frac{4}{\lambda} - 1 \Rightarrow \sigma^2_{\text{FAPO}} < \sigma^2_{\text{GRPO}} \\
&\text{when } \frac{\alpha}{\beta} > \frac{4}{\lambda} - 1 \Rightarrow \sigma^2_{\text{FAPO}} > \sigma^2_{\text{GRPO}}
\end{aligned}
\tag{18}
$$

**Summary: The Whole Optimization Process of FAPO**   We introduce $\rho = \frac{\alpha}{\beta}$ to characterize the current optimization state, which increases monotonically from $0$. For a pre-defined and fixed $\lambda$, the learning process drives $\rho$ upward. Once $\rho$ exceeds $\frac{2}{\lambda} - 1$, the optimization direction shifts from reaching the correct answer (warm-up stage) toward reinforcing reliable reasoning (refinement stage). As $\rho$ continues to increase and surpasses $\frac{4}{\lambda} - 1$, the scaling factor $\sigma$ rises accordingly, making the advantage estimation more conservative (i.e., $|\hat{A}_{\text{FAPO}}| < |\hat{A}_{\text{GRPO}}|$, for flawed-positive and fully correct rollouts). This conservativeness helps to stabilize training by preventing overly aggressive updates, while still ensuring that correct and reliable rollouts are consistently prioritized.

**Determining** $\lambda$  From the above analysis, the reward assignment parameter $\lambda$ plays a central role in determining when the optimization shift is applied, whether conservative or aggressive. In practice, we adopt a majority-guided strategy, which provides both intuitiveness and effectiveness. In the early stage of training, when negative samples with proportion $\beta$ dominate, flawed positives are assigned positive advantages, enabling the model to acquire the ability to produce correct answers quickly. As training progresses and fully correct rollouts become the majority (i.e., $\alpha > \beta$), the optimization naturally shifts toward reinforcing reliable reasoning. Formally:

$$\rho_{\text{shift}} = \frac{\alpha}{\beta} = 1 \Rightarrow \lambda = \frac{2}{\rho_{\text{shift}} + 1} = 1. \tag{19}$$

Therefore, we set $\lambda = 1$ as the default configuration in FAPO.

## B  IMPLEMENTATION DETAILS

**FAPO-Reasoning**  Table 1 summarizes the training configurations and hyperparameters of our generative reward model (GenRM) and final reasoning models. For GenRM training, we follow the practice of An et al. (2025) by using a higher rollout temperature to encourage exploration. For reasoning model training, most settings are consistent with DAPO (Yu et al., 2025), except that we reduce the number of rollouts from 16 to 8 to accelerate overall training speed.

Table 1: Training configurations and hyperparameters of our experiments.

| | FAPO-GenRM-4B | Baseline & FAPO-7B | Baseline & FAPO-32B |
|---|---|---|---|
| **Data Configuration** | | | |
| Global Batch Size | 512 | 512 | 512 |
| Base Model | Qwen3-4B-Instruct | Qwen2.5-Math-7B | Qwen2.5-32B |
| **Rollout Inference** | | | |
| Rollout Num per Prompt | 16 | 8 | 8 |
| Temperature | 1.2 | 1.0 | 1.0 |
| Top-p | 1.0 | 1.0 | 1.0 |
| Top-k | -1 | -1 | -1 |
| Max Prompt Length | 5120 | 2048 | 2048 |
| Max Response Length | 8192 | 8192 | 20480 |
| Overlong Buffer Length | 4096 | 4096 | 4096 |
| Overlong Penalty Factor | 1.0 | 1.0 | 1.0 |
| **Actor Training** | | | |
| PPO Mini Batch Size | 32 | 32 | 32 |
| Advantage Estimation Type | GRPO | GRPO | GRPO |
| Clipping $\epsilon_{\text{low}}$ | 0.2 | 0.2 | 0.2 |
| Clipping $\epsilon_{\text{high}}$ | 0.28 | 0.28 | 0.28 |
| Optimizer | Adam | Adam | Adam |
| Learning Rate | $10^{-6}$ | $10^{-6}$ | $10^{-6}$ |
| Weight Decay | 0.1 | 0.1 | 0.1 |
| $(\beta_1, \beta_2)$ | $(0.9, 0.999)$ | $(0.9, 0.999)$ | $(0.9, 0.999)$ |
| Gradient norm clipping | 1.0 | 1.0 | 1.0 |
| Learning Rate Scheduler | constant | constant | constant |
| Warmup Steps | 10 | 10 | 10 |
| **Evaluation Setup** | | | |
| Temperature | 0.6 | 1.0 | 1.0 |
| Top-p | 0.95 | 0.7 | 0.7 |
| Top-k | -1 | -1 | -1 |
| Max Generation Length | 8192 | 8192 | 20480 |

**FAPO-GenRM**  FAPO-GenRM is trained using the FAPO-critic dataset, where the ground truth label is generated by Qwen-32B. To mitigate the label noise in the data and train FAPO-GenRM robustly, we propose the following strategies:

**Overview of Agent Reward Loop**

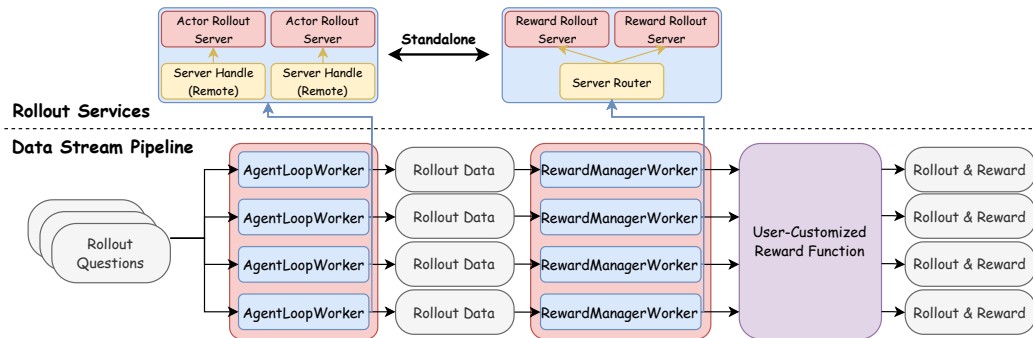

Figure 9: Infrastructure Design of Reward Loop

- **Consensus filtering:** During data synthesis, we sample each instance three times. A sample is kept only if all three generations yield consistent outcomes (i.e., the process errors occur at exactly the same positions). As a result, the retained samples tend to be highly reliable and have strong internal agreement.
- **Robust Training Objective:** Our designed training objective (in Equation 7) are robust to the subtle errors missed by the teacher. The reward supervision signal is a soft noise-robust label, the student can still receive an appropriate reward even when the annotated error location deviates slightly from the true error.

**Reward Loop Design**    Figure 9 shows the infrastructure design of the reward loop, a partially asynchronous infrastructure for reward computation. Reward Loop is designed for:

- **Make reward computation more efficient:** each rollout sample is sent to the reward model immediately after it is generated, without waiting for the full batch rollouts to finish.
- **Make user customized reward function more flexible:** the design of the Reward Loop provides substantial flexibility for implementing custom reward functions and supports both DiSRM and GenRM.

This implementation is based on the veRL agent loop (Sheng et al., 2025) and has been open-sourced (relevant implementation can be checked in our supplementary materials).

## C  ADDITIONAL RESULTS AND DISCUSSION

**Additional Results of FAPO-GenRM**    Table 2 shows a detailed comparison of our FAPO-GenRM-4B model with other publicly available discriminative and generative reward models. Building upon Qwen3-4B-Instruct, our model achieves substantial improvements in both Flawed-PositiveBench and ProcessBench (Zheng et al., 2024), even surpassing Qwen3-32B and the discriminative SoTA model Qwen2.5-Math-PRM-72B (Zhang et al., 2025b).

**Additional Results of FAPO-Reasoning**    Strong generalization ability is one of the key advantages of reinforcement learning (Huan et al., 2025). We then extend the evaluation of FAPO to additional tasks, including two more mathematical domains (MATH (Hendrycks et al., 2021) and AMC) and code reasoning (LiveCodeBench (Jain et al., 2024)). The overall results are shown in the Table 3. FAPO outperforms the baseline in code reasoning tasks and achieves consistent improvements across a broad range of tasks.

**FAPO application in large-scale RL systems**    Introducing an extra generative reward model will bring a burden to the systems. We then quantify the burden and explore the potential application of FAPO in large-scale RL systems. We provide a more detailed breakdown of the time distribution across different RL stages in Table 4.

Table 2: FAPO-GenRM results in FlawedPositiveBench and ProcessBench.

| Model | FlawedPositiveBench | | | ProcessBench |
|---|---|---|---|---|
| | Precision | Recall | $F_1$ | Avg. $F_1$ |
| **Discriminative Process Models (7B-70B)** | | | | |
| Qwen2.5-Math-PRM-7B | 76.6 | 83.5 | 79.9 | 70.5 |
| Qwen2.5-Math-PRM-72B | 74.3 | 91.0 | 81.8 | 76.8 |
| **Generative Critic Models (7B-70B)** | | | | |
| Qwen2.5-Math-7B-Instruct | 58.0 | 43.8 | 49.9 | 19.9 |
| Qwen2.5-7B-Instruct | 50.0 | 66.2 | 57.0 | 38.9 |
| Qwen3-1.7B (Think) | 73.4 | 75.1 | 74.2 | 56.0 |
| Qwen3-4B-Instruct | 69.4 | 97.9 | 81.2 | 77.1 |
| Qwen3-4B (Think) | 89.2 | 84.3 | 82.0 | 79.6 |
| Qwen3-32B (Think) | 85.1 | 90.8 | 87.8 | 82.0 |
| **FAPO-GenRM-4B (Ours)** | 87.5 | 91.4 | **89.4** | **83.3** |

Table 3: FAPO-Reasoning results in more evaluation benchmarks.

| Model | Math | | | | Code | General | Avg. |
|---|---|---|---|---|---|---|---|
| | AIME24 | AIME25 | AMC | MATH | LiveCodeBench | GPQA-Diamond | |
| Baseline-32B | 38.9 | 29.5 | 85.0 | 72.8 | 28.6 | 51.0 | 51.0 |
| FAPO-32B | **42.4** | **33.5** | **91.6** | **74.6** | **33.6** | **53.1** | **54.8** |

The relative inference cost for long-cot models actually decreases, specifically: (1) Rollout: Long-CoT models exhibit a strong long-tail issue: the generation time is bounded by the longest trajectory in the batch. Therefore, rollout time increases significantly as the model generates longer traces (e.g., 12k tokens for Qwen3-4B). (2)GenRM: In contrast, GenRM inference time does not increase with longer trajectories. We observe that the GenRM output length stays nearly constant across models (e.g., $3.2k \rightarrow 3.8k$). (3) Policy update: This stage accounts for a relatively small portion of the total cost and primarily scales with model size rather than trajectory length. Overall, the root cause is that the critic task for FAPO-GenRM does not require a very long response length.

**Ablation study of hyper-parameter $\lambda$**    $\lambda$ is the only key hyper-parameter introduced in the FAPO algorithm. We explore the parameter tuning based on the FAPO-7B setting, with results illustrated in Table 5

We can conclude that:

- Flaw-aware learning consistently improves performance, as all configurations outperform the baseline.
- Achieving the best performance require tuning the parameter $\lambda$. In the 7B setting, a more aggressive strategy leads to larger gains. The configuration with $\rho = 1/2$, which corresponds to an optimization shift where about one third of the rollouts are fully correct, achieves the best performance.

So overall, flaw-aware learning leads to performance gains, but achieving the best performance requires tuning the parameter $\lambda$. That said, FAPO introduces only this single additional parameter, which makes the tuning process relatively easy.

**Human Verification on the reliability of FAPO-32B**    Beyond the previous LLM-as-a-judge evaluation with Qwen3-32B, we further conduct a detailed human verification to assess the process reliability of FAPO-32B. Specifically, we randomly sample 20 positive cases with correct final answers and manually examine whether they contain unreliable reasoning patterns, with the results summarized in Table 6. Through this analysis, we observe that (1) multiple-choice questions exhibit a noticeably higher proportion of flawed positives compared to math word problems, where the final answer is often a symbolic expression; and (2) our method consistently outperforms the

Table 4: Time distribution across different RL stages in different settings.

| Model | Rollout (Infer) | Rollout Len | FAPO-GenRM (Infer) | GenRM Len | Policy Update (Train) | Nodes |
|---|---|---|---|---|---|---|
| FAPO-7B | 42% | 1.1k | **18%** | 3.0k | 33% | **4** |
| FAPO-32B | 60% | 2.3k | **14%** | 3.2k | 20% | **8** |
| Qwen3-4B | 72% | 12.0k | **10%** | 3.8k | 14% | **16** |

Table 5: Hyper-parameter tuning of $\lambda$.

| Setting | Performance |
|---|---|
| Baseline-7B ($\rho = +\infty \Rightarrow \lambda = 0$) | 32.1 |
| FAPO with $\rho = 2 \Rightarrow \lambda = 1/3$ | 34.6 |
| FAPO with $\rho = 1 \Rightarrow \lambda = 1$ (default setting) | 36.8 |
| FAPO with $\rho = 1/2 \Rightarrow \lambda = -1/3$ | 39.6 |

baseline under both LLM-as-a-judge and human verification, further demonstrating its effectiveness in mitigating flawed-positive issues.

Table 6: LLM-as-a-judge and human verification of flawed positive ratio.

| Model | AIME24 (Math Word Problem) | AIME25 (Math Word Problem) | GPQA-Diamond (Multi-Choice Problem) |
|---|---|---|---|
| Baseline-32B (LLM-as-a-judge) | 15.5 | 10.9 | 45.7 |
| FAPO-32B (LLM-as-a-judge) | **7.1 (-8.4)** | **1.7 (-9.2)** | **42.0 (-3.7)** |
| Baseline-32B (Human-Verification) | 4 / 20 | 1 / 20 | 10 / 20 |
| FAPO-32B (Human-Verification) | **2 / 20** | **0 / 20** | **7 / 20** |

**How Flawed Learning contributes to Performance Gains** In section 2.2, we demonstrate that the flawed positive acts as a role of stepping stones to performance gains, and Figure 2 (d) illustrates the correlation between them. Here, we provide a more complete explanation of how mitigating flawed positives leads to performance gains.

- **Early Stages: Flawed Behavior → More Correct Rollouts:** Prior studies (Zheng et al., 2024; Wang et al., 2025) have shown that flawed reasoning often acts as a shortcut to the correct final answer. This bias, inherited from pre-training (Kalai et al., 2025), leads to certain flawed rollouts. Thus, in the early phase of RL, flawed positives naturally increase the number of correct final answer rollouts.
- **Early Stages: More Correct Rollouts → Early performance gains:** A larger pool of correct rollouts yields more positive rewards, providing stronger supervision and driving exploitation early in training. This effect can be directly observed in train-time reward statistics in Table 7.
- **Later Stages: Penalizing flawed behavior → Fewer flawed rollouts:** Our theoretical analysis demonstrates the optimization distribution shift: flawed rollouts receive negative advantage, progressively reducing the model's tendency to produce flawed processes. This trend is reflected in Figure 1 (left), where the proportion of flawed positives decreases steadily as training progresses.
- **Later Stages: Fewer flawed rollouts → Performance gains:** As flawed rollouts diminish, the model allocates more rollout chances to fully correct trajectories. Consequently, the RL loop receives a higher density of genuinely useful reward signals, as reflected in Table 8, improving final performance.

**Model Selection in FAPO Experiments** We explain the model selection in both FAPO-GenRM and FAPO-Reasoning:

Table 7: Early-stage train-time rewards (view flawed rollout as positive rollout).

| Step | 10 | 20 | 30 | 40 | 50 |
|---|---|---|---|---|---|
| Baseline-7B | 0.14 | 0.27 | 0.29 | 0.34 | 0.36 |
| FAPO-7B | **0.18** | **0.33** | **0.37** | **0.42** | **0.44** |

Table 8: Later-stage train-time rewards (view flawed rollout as negative rollout).

| Step | 120 | 140 | 160 | 180 | 200 |
|---|---|---|---|---|---|
| Baseline-7B | 0.35 | 0.37 | 0.36 | 0.40 | 0.39 |
| FAPO-7B | **0.38** | **0.40** | **0.41** | **0.44** | **0.45** |

- **Model Selection in FAPO-GenRM:** We adopt Qwen3-4B-Instruct as the base model for GenRM training, considering three factors: (1) it demonstrates strong instruction-following and basic error-detection capability (as shown in Table 2), making it a suitable initialization for RL; (2) its relatively small size ensures efficient training and faster inference, with Instruct models producing shorter responses than think-style models; (3) it avoids potential concerns of knowledge leakage into the final RL process that may arise if the base model is overly strong.
- **Model Selection in FAPO-Reasoning:** We follow the same setup as DAPO (Yu et al., 2025) and use Qwen2.5-Math-7B and Qwen2.5-32B as base models, as (1) the learning curves of pre-trained models clearly illustrate the entire exploration–exploitation trajectory starting from near-zero performance, whereas RL-finetuned Instruct models often exhibit instability; (2) these two models also strike a good balance between response length and training speed.

## D  LIMITATIONS AND FUTURE WORK

Our work presents several limitations that point to promising future directions, both in algorithmic effectiveness and infrastructure design. On the algorithmic side, although this work trains on mathematical reasoning tasks, FAPO has strong potential in broader settings such as multi-choice tasks, multi-turn interactions, and agent-based RL, where flawed processes are often more pronounced and problematic. We will also further validate the effectiveness of FAPO across a wider range of model architectures (e.g., MoE) and larger model scales. On the infrastructure side, while our decoupled design improves efficiency and inference speed, its applicability to fully asynchronous RL systems remains uncertain, as we discussed in Section 4.5, specifically Table 1. We regard these as important research directions in our future work.

## E  LLM USAGE

During the writing of this paper, AI assistants are employed to assist with minor language refinement. Their suggestions are limited to enhancing clarity and readability, without influencing the research design, experiments, or conclusions. All content was carefully reviewed, validated, and revised by the authors to ensure accuracy and fidelity to the research.

## F  PROMPTS

> **Prompts for Flawed Positive Detection**
>
> **Prompt for Outcome Reward Model (ORM):**
> The following is a math problem with its ground truth answer, along with an AI solution:
>
> **[Math Problem]**
> {problem statement}
>
> **[Ground Truth]**
> {ground truth answer}
>
> **[AI Solution]**
> {AI Solution}

```
Your task is to review and critique the solution step by step, and
output whether the AI solution is correct.
Please reason step by step, put your final answer (i.e., 'True' or
'False') in \boxed{}
```

**Prompt for Process Reward Model (PRM):**
The following is a math problem with its ground truth answer, along
with an AI solution (split into paragraphs, enclosed with tags and
indexed from 0):

**[Math Problem]**
{*problem statement*}

**[Ground Truth]**
{*ground truth answer*}

**[AI Solution]**
```
<paragraph_0>...</paragraph_0>
<paragraph_1>...</paragraph_1>
......
<paragraph_(n-1)>...</paragraph_(n-1)>
```

Your task is to review and critique the solution paragraph by
paragraph. Once you identify an error in a paragraph, return the
index of the paragraph where the earliest error occurs. Otherwise,
return the index of -1 (which typically denotes 'not found').
Please reason step by step, put your final answer (i.e., the index)
in \boxed{}

---

### Prompts for Reasoning

**Prompt for DAPO-Math-17K, AIME24, and AIME25:**
{*problem statement*}

Please reason step by step, and put your final answer within
\boxed{}.

**Prompt for GPQA-Diamond:**
{*problem statement*}
A. {*Choice_A*}
B. {*Choice_B*}
C. {*Choice_C*}
D. {*Choice_D*}

Please reason step by step, and put your final answer (only the
choice letter) within \boxed{}.

---

## G FLAWED POSITIVE SAMPLES WITH HUMAN ANALYSIS

### Flawed Positive Example 1

**Question Source: AIME24**
**Question:**
Alice chooses a set $A$ of positive integers. Then Bob lists all
finite nonempty sets $B$ of positive integers with the property that
the maximum element of $B$ belongs to $A$. Bob's list has 2024 sets.
Find the sum of the elements of A.

---

**Generation Model: Qwen2.5-Math-7B (Pre-Trained Base Model)**
**Response:**
We are given that Bob lists all finite nonempty sets $B$ of positive

integers where the maximum element of $B$ belongs to $A$. Bob's list
contains 2024 such sets. We need to find the sum of the elements of
$A$.

Step-by-Step Reasoning:

1. **Understanding the Problem:**
- Let $A = \{a_1, a_2, \ldots, a_n\}$ where $a_1 < a_2 < \ldots < a_n$.
- For each $a_i \in A$, Bob lists all finite nonempty sets $B$ such that
the maximum element of $B$ is $a_i$.

2. **Counting the Sets for Each $a_i$:**
- For a given $a_i$, the sets $B$ can include any subset of the integers
less than $a_i$, plus $a_i$ itself.
- The number of subsets of the integers less than $a_i$ is $2^{a_i - 1}$ (since
there are $a_i - 1$ integers less than $a_i$).
- However, since $B$ must be nonempty, we subtract 1 from this count
(the empty set).

**Therefore, the number of sets $B$ for a given $a_i$ is:**

$$2^{a_i - 1} - 1$$

3. **Total Number of Sets:**
- The total number of sets listed by Bob is the sum of the number of
sets for each $a_i$:

$$\sum_{i=1}^{n} (2^{a_i - 1} - 1) = 2024$$

- Simplifying, we get:

$$\sum_{i=1}^{n} 2^{a_i - 1} - n = 2024$$

$$\sum_{i=1}^{n} 2^{a_i - 1} = 2024 + n$$

4. **Finding $n$:**
- We need to find $n$ such that the sum $\sum_{i=1}^{n} 2^{a_i - 1}$ is minimized.
- The smallest possible values for $a_i$ are $1, 2, 3, \ldots, n$.

Let's test with small values of $n$:
- For $n = 1$:

$$2^{1-1} = 1 \quad \text{(too small)}$$

- For $n = 2$:

$$2^{1-1} + 2^{2-1} = 1 + 2 = 3 \quad \text{(too small)}$$

- For $n = 3$:

$$2^{1-1} + 2^{2-1} + 2^{3-1} = 1 + 2 + 4 = 7 \quad \text{(too small)}$$

- For $n = 4$:

$$2^{1-1} + 2^{2-1} + 2^{3-1} + 2^{4-1} = 1 + 2 + 4 + 8 = 15 \quad \text{(too small)}$$

- For $n = 5$:

$$2^{1-1} + 2^{2-1} + 2^{3-1} + 2^{4-1} + 2^{5-1} = 1 + 2 + 4 + 8 + 16 = 31 \quad \text{(too small)}$$

- For $n = 6$:

$$2^{1-1} + 2^{2-1} + 2^{3-1} + 2^{4-1} + 2^{5-1} + 2^{6-1} = 1 + 2 + 4 + 8 + 16 + 32 = 63 \quad \text{(too small)}$$

- For $n = 7$:

$$2^{1-1} + 2^{2-1} + 2^{3-1} + 2^{4-1} + 2^{5-1} + 2^{6-1} + 2^{7-1} = 127 \quad \text{(too small)}$$

- For $n = 8$:

$$2^{1-1} + 2^{2-1} + 2^{3-1} + 2^{4-1} + 2^{5-1} + 2^{6-1} + 2^{7-1} + 2^{8-1} = 255 \quad \text{(too small)}$$

- For $n = 9$:

$$2^{1-1} + 2^{2-1} + 2^{3-1} + 2^{4-1} + 2^{5-1} + 2^{6-1} + 2^{7-1} + 2^{8-1} + 2^{9-1} = 511 \quad \text{(too small)}$$

- For $n = 10$:

$2^{1-1}+2^{2-1}+2^{3-1}+2^{4-1}+2^{5-1}+2^{6-1}+2^{7-1}+2^{8-1}+2^{9-1}+2^{10-1} = 1023$ (too small)

- For $n = 11$:

$2^{1-1}+2^{2-1}+2^{3-1}+2^{4-1}+2^{5-1}+2^{6-1}+2^{7-1}+2^{8-1}+2^{9-1}+2^{10-1}+2^{11-1} = 2047$ (too large)

Therefore, $n = 10$ is the correct value because:

$$2^{1-1} + 2^{2-1} + 2^{3-1} + 2^{4-1} + 2^{5-1} + 2^{6-1} + 2^{7-1} + 2^{8-1} + 2^{9-1} + 2^{10-1} = 1023$$

And:
$$1023 + 10 = 1033 \quad \text{(which is too large)}$$

So, we need to adjust $n$ to be 10 and check the sum:

$$2^{1-1} + 2^{2-1} + 2^{3-1} + 2^{4-1} + 2^{5-1} + 2^{6-1} + 2^{7-1} + 2^{8-1} + 2^{9-1} + 2^{10-1} = 1023$$

And:
$$1023 + 10 = 1033 \quad \text{(which is too large)}$$

**Therefore, the correct $n$ is 10, and the sum of the elements of $A$ is:**
$$1 + 2 + 3 + 4 + 5 + 6 + 7 + 8 + 9 + 10 = 55$$

**Thus, the sum of the elements of $A$ is:**

$$\boxed{55}$$

---

**Human Analysis:** There are two clear mistakes in the AI's response:

1. The number of sets $B$ for a $a_i$ should be $2^{a_i-1}$ (not $2^{a_i-1} - 1$). This incorrect assumption leads to a completely flawed derivation throughout the rest of the solution.

2. The correct set $A$ should be $\{4, 6, 7, 8, 9, 10, 11\}$, instead of $\{1, 2, 3, 4, 5, 6, 7, 8, 9, 10\}$. From the solution process, it's clear that the AI guessed a value close to the target, rather than identifying the exact condition required.

**In summary, although the final answer happens to be correct, the overall reasoning process is fundamentally flawed.**

---

## Flawed Positive Example 2

**Question Source:** DAPO-Math-17K
**Question:**
Let $a$, $b$, $c$ be the three roots of the equation $x^3 - (k+1)x^2 + kx + 12 = 0$, where $k$ is a real number. If $(a-2)^3 + (b-2)^3 + (c-2)^3 = -18$, find the value of $k$.

---

**Response Generation Model:** Llama-3.3-70B-Instruct (ShortCoT Model)
**Response:**
**Step 1: Recall Vieta's formulas for a cubic equation**
Vieta's formulas for a cubic equation $x^3 + px^2 + qx + r = 0$ with roots $a$, $b$, and $c$ state that: $a + b + c = -p$, $ab + bc + ac = q$, and $abc = -r$. Applying these formulas to the given equation $x^3 - (k+1)x^2 + kx + 12 = 0$, we get $a + b + c = k + 1$, $ab + bc + ac = k$, and $abc = -12$.
**Step 2: Expand the given expression using the identity for $(a-2)^3 + (b-2)^3 + (c-2)^3$**
The identity for $(a - 2)^3 + (b - 2)^3 + (c - 2)^3$ can be expanded as $(a^3 - 6a^2 + 12a - 8) + (b^3 - 6b^2 + 12b - 8) + (c^3 - 6c^2 + 12c - 8)$. This simplifies to $(a^3 + b^3 + c^3) - 6(a^2 + b^2 + c^2) + 12(a + b + c) - 24$.
**Step 3: Use the given equation to find $a^3 + b^3 + c^3$**
Since $a$, $b$, and $c$ are roots of the equation $x^3 - (k+1)x^2 + kx + 12 = 0$, we can express $a^3$, $b^3$, and $c^3$ in terms of $a$, $b$, $c$, and $k$. Substituting

$x = a$, $b$, and $c$ into the equation gives $a^3 = (k + 1)a^2 - ka - 12$, $b^3 = (k+1)b^2 - kb - 12$, and $c^3 = (k+1)c^2 - kc - 12$. Summing these gives $a^3 + b^3 + c^3 = (k+1)(a^2 + b^2 + c^2) - k(a+b+c) - 36$.

**Step 4: Find $a^2 + b^2 + c^2$ using the sum of roots**
From Vieta's formulas, we know $a + b + c = k + 1$. Squaring both sides gives $(a + b + c)^2 = (k + 1)^2$, which expands to $a^2 + b^2 + c^2 + 2(ab + bc + ac) = k^2 + 2k + 1$. Since $ab + bc + ac = k$, we can substitute this in to get $a^2 + b^2 + c^2 + 2k = k^2 + 2k + 1$, simplifying to $a^2 + b^2 + c^2 = k^2 + 1$.

**Step 5: Substitute $a^2 + b^2 + c^2$ into the expression for $a^3 + b^3 + c^3$**
Substituting $a^2 + b^2 + c^2 = k^2 + 1$ into $a^3 + b^3 + c^3 = (k+1)(a^2 + b^2 + c^2) - k(a+b+c) - 36$ gives $a^3 + b^3 + c^3 = (k + 1)(k^2 + 1) - k(k + 1) - 36$. Simplifying yields $a^3 + b^3 + c^3 = k^3 + k + k^2 + 1 - k^2 - k - 36 = k^3 - 35$.

**Step 6: Substitute known values into the expanded expression for $(a-2)^3 + (b-2)^3 + (c-2)^3$**
Given that $(a - 2)^3 + (b - 2)^3 + (c - 2)^3 = -18$, and using the expansion from Step 2, we substitute $a^3 + b^3 + c^3 = k^3 - 35$ and $a + b + c = k + 1$ into the expression. We also need $a^2 + b^2 + c^2 = k^2 + 1$ from Step 4. The expression becomes $(k^3 - 35) - 6(k^2 + 1) + 12(k + 1) - 24 = -18$.

**Step 7: Solve for $k$**
**Simplifying the equation** $(k^3 - 35) - 6(k^2 + 1) + 12(k + 1) - 24 = -18$ **gives** $k^3 - 6k^2 + 12k - 35 - 6 + 12k + 12 - 24 = -18$. **Combining like terms yields** $k^3 - 6k^2 + 24k - 53 = -18$. **Adding 18 to both sides gives** $k^3 - 6k^2 + 24k - 35 = 0$. This is a cubic equation in terms of $k$.

**Step 8: Solve the cubic equation for $k$**
To solve $k^3 - 6k^2 + 24k - 35 = 0$, let's try to factor it or find a rational root. Checking for simple integer solutions, $k = 5$ is found to be a root (through trial and error or using the Rational Root Theorem), as $5^3 - 6(5)^2 + 24(5) - 35 = 125 - 150 + 120 - 35 = 60$. However, this was an error in calculation: the correct calculation for $k = 5$ should be $5^3 - 6(5)^2 + 24(5) - 35 = 125 - 150 + 120 - 35 = 60$, which is not correct. The correct approach is to actually solve the equation correctly. Let's correct the step: We need to solve $k^3 - 6k^2 + 24k - 35 = 0$. Trying $k = 5$ correctly: $5^3 - 6(5)^2 + 24(5) - 35 = 125 - 150 + 120 - 35 = 60$, which does not equal 0, indicating an error in identifying $k = 5$ as a root. The correct factorization or solution method must be applied.

**Step 9: Correctly identify and solve for $k$**
The error in Step 8 indicates a miscalculation in identifying a root. The correct approach involves factoring or applying a method like synthetic division if a rational root is suspected, or using numerical methods. However, given the mistake in calculation and approach, let's reconsider the equation and directly evaluate potential rational roots or use given conditions more effectively.

**The final answer is:** $\boxed{5}$

---

**Human Analysis:** The correct simplified equation is $k^3 - 6k^2 + 12k - 35 = 0$, not $k^3 - 6k^2 + 24k - 35 = 0$. This typo leads to an incorrect solution of $k$. **However, what's quite peculiar is that despite the model's multiple failed attempts to correct itself, it ultimately guessed $k = 5$.**

---

## Flawed Positive Example 3

**Question Source: DAPO-Math-17K**
**Question:**
Let $a, b, c$ be distinct numbers such that the equations $x^2 + ax + 1 = 0$ and $x^2 + bx + c = 0$ have a common real root, and the equations $x^2 + x + a = 0$ and $x^2 + cx + b = 0$ also have a common real root. Compute the sum $a + b + c$.

**Response Generation Model: Qwen3-1.7B (LongCoT Model, Think Mode)**
**Response:**
`<think>`...`</think>` (Think Content Omitted Here)

To solve the problem, we are given three distinct numbers $a, b, c$ such that:
- The equations $x^2 + ax + 1 = 0$ and $x^2 + bx + c = 0$ share a common real root,
- The equations $x^2 + x + a = 0$ and $x^2 + cx + b = 0$ also share a common real root.

We are to compute the value of $a + b + c$.

---

**Step 1: Use the Common Root Condition**

Let $\alpha$ be a real root of both equations $x^2 + ax + 1 = 0$ and $x^2 + bx + c = 0$. Substituting $\alpha$:

$$\alpha^2 + a\alpha + 1 = 0 \quad (1) \quad \alpha^2 + b\alpha + c = 0 \quad (2)$$

Subtracting (1) from (2):

$$(b - a)\alpha + (c - 1) = 0 \Rightarrow \alpha = \frac{1 - c}{b - a} \quad (3)$$

Now, substitute $\alpha$ into (1) and (2) to express relationships between $a, b, c$.

---

**Step 2: Use the Second Pair of Equations**

Let $\beta$ be a real root of both $x^2 + x + a = 0$ and $x^2 + cx + b = 0$. Substituting $\beta$:

$$\beta^2 + \beta + a = 0 \quad (4) \quad \beta^2 + c\beta + b = 0 \quad (5)$$

Subtracting (4) from (5):

$$(c - 1)\beta + (b - a) = 0 \Rightarrow \beta = \frac{a - b}{c - 1} \quad (6)$$

---

**Step 3: Use the Relationship Between $\alpha$ and $\beta$**

From the problem, it is known that $\beta = \frac{1}{\alpha}$. Therefore:

$$\frac{a - b}{c - 1} = \frac{1}{\alpha} \Rightarrow \alpha = \frac{1 - c}{b - a}$$

Also, from (6):

$$\beta = \frac{a - b}{c - 1}$$

So:

$$\beta = \frac{a - b}{c - 1} = \frac{1}{\alpha} = \frac{1 - c}{b - a}$$

This gives a relationship between $a, b, c$, **and we proceed by assuming $\alpha = 1$, which simplifies the algebra.**

---

**Step 4: Find $a, b, c$ with $\alpha = 1$**

Suppose $\alpha = 1$ is a common root of both pairs of equations. From equation (1):

$$1^2 + a \cdot 1 + 1 = 0 \Rightarrow 1 + a + 1 = 0 \Rightarrow a = -2$$

Now, since $\alpha = 1$ is a root of equation (2):

$$1^2 + b \cdot 1 + c = 0 \Rightarrow 1 + b + c = 0 \Rightarrow c = -1 - b$$

Now, from the second pair of equations, $\beta = 1$ is a root of both $x^2 + x + a = 0$ and $x^2 + cx + b = 0$. Substitute $x = 1$ into both equations:
- $1^2 + 1 + a = 0 \Rightarrow 1 + 1 + (-2) = 0 \Rightarrow 0$ ✓
- $1^2 + c \cdot 1 + b = 0 \Rightarrow 1 + c + b = 0 \Rightarrow c + b = -1$ ✓

Thus, consistent.

Now, the sum:

$$a + b + c = -2 + b + (-1 - b) = -3$$

---

**Final Answer**

$$\boxed{-3}$$

---

**Human Analysis:** The AI makes an unwarranted assumption that $\alpha = 1$, whereas in fact, deriving this requires a complex reasoning process. Although the final answer is correct, the reasoning is flawed and the correct value is reached coincidentally.

