# OpenReview forum: "FAPO: Flawed-Aware Policy Optimization for Efficient and Reliable Reasoning"
_ICLR.cc/2026/Conference — ICLR 2026 Poster_

### Official Review · Reviewer_Gwxh · 2025-10-31

**Soundness:** 3
**Presentation:** 3
**Contribution:** 2
**Rating:** 4
**Confidence:** 4

**Summary:**

This paper introduces FAPO, a novel approach to improving RL for LLMs in reasoning tasks. It identifies that flawed-positive rollouts persist throughout RL training and hinder model performance, then proposes FAPO which combines a compact generative reward model (FAPO-GenRM-4B) to detect reasoning errors with process-level rewards and a parameter-free algorithm that applies adaptive penalties to flawed positives. FAPO demonstrates consistent improvements across 7B and 32B models on mathematical (AIME24: +4.7%, AIME25: +3.1%) and general reasoning tasks (GPQA-Diamond: +1.5%), while substantially reducing flawed-positive ratios.

**Strengths:**

1. The paper is well-written and easy to follow. In particular, figures in the paper effectively illustrate key concepts
2. The systematic study in Section 2.2 provides valuable insights into the prevalence and evolution of flawed-positive rollouts during RL training, supported by both automatic evaluation and human verification
3. The effectiveness of FAPO is demonstrated both theoretically and empricially

**Weaknesses:**

1. The hybrid process + outcome reward formulation is not novel. Prior works have extensively explored combining step-level and outcome-level rewards (e.g., arXiv:2312.08935, arXiv:2504.13958)
2. The paper only evaluates on mathematical reasoning tasks (AIME24, 25) and one general domain benchmark (GPQA-Diamond), which is insufficient to claim broad applicability. The paper would benefit significantly from including additional widely-used benchmarks such as AMC and MATH
3. Only two model families (Qwen2.5-Math-7B and Qwen2.5-32B) are tested, both starting from pre-trained models. It would be valuable to stress-test the proposed GenRM on models that already have some code-starts with long-CoT trajectories
4. While the improvements are consistent, the absolute performance gains come at considerable cost: (1) requires training an additional 4B reward model, (2) needs complex asynchronous infrastructure

**Questions:**

1. Why does the student model outperform the teacher model? Table 2 and Figure 3 show that FAPO-GenRM-4B outperforms Qwen3-32B (teacher) on FlawedPositiveBench
2. What is the overall computational cost overhead? The paper mentions training time increases by "less than 20%" but lacks comprehensive cost analysis (e.g., the cost of data synthesis, training GenRM, and detailed inference costs during RL). In particular, I am curious whether the inference cost would increase for GenRM if performing FAPO on models that have some code-starts with long-CoT

---

> ### Author Response · Authors · 2025-11-22
> **Rebuttal by Authors [1/N]**
>
> Thank you for your feedback and the opportunity to clarify our contributions and positioning.
>
> > W1: The hybrid process + outcome reward formulation is not novel. Prior works have extensively explored combining step-level and outcome-level rewards (e.g., arXiv:2312.08935, arXiv:2504.13958)
>
>
> We agree with the reviewer that process supervision has long been recognized as a promising direction for AI alignment and LLM reasoning, and numerous studies have reported encouraging results.
>
> However, existing work primarily focus on discriminative reward models [1,2] and rule-based process supervision [3]. While generative rewards models have demonstrated strong potential [4] in providing interpretable textual feedback, their application in RL remains very limited.
> FAPO introduces a new paradigm of generative reward modeling (GenRM) for RL, which differs substantially from prior work and brings several key contributions:
> - **Algorithmic contribution:** FAPO is, to our knowledge, **the first to bridge and open-source** the full pipeline from training to deploying GRMs effectively in RL. Our results show that GRM can provide strong supervision signals for policy optimization, be robust to reward hacking and bring substantial improvements.
> - **Open-source infrastructure and resources:** The community currently lacks practical frameworks for RL with GenRMs. FAPO introduces an efficient and easy-to-use **Reward Loop Infrastructure**, providing an important foundation for future GRM research.
>
> Overall, FAPO **offers a new GenRM paradigm for RL (including effective algorithm and usable infrastructures)**, providing richer supervision and improved robustness, which we believe meaningfully distinguishes our approach from previous works.
>
> [1] Math-Shepherd: Verify and Reinforce LLMs Step-by-step without Human Annotations
>
> [2] Process Reinforcement through Implicit Rewards
>
> [3] ToolRL: Reward is All Tool Learning Needs
>
> [4] Inference-Time Scaling for Generalist Reward Modeling
>
> > W2: The paper only evaluates on mathematical reasoning tasks (AIME24, 25) and one general domain benchmark (GPQA-Diamond), which is insufficient to claim broad applicability. The paper would benefit significantly from including additional widely-used benchmarks such as AMC and MATH
>
> This is a crucial point. We have extend to two more math tasks (amc and math) and one more code reasoning tasks (livecodebench) as below:
>
> |              | AIME24   | AIME25   | AMC      | MATH     | **LiveCodeBench** | GPQA-Diamond | Avg             |
> | ------------ | -------- | -------- | -------- | -------- | ----------------- | ------------ | --------------- |
> | Baseline-32B | 38.9     | 29.5     | 85.0     | 72.8     | 28.6              | 51.0         | 51.0            |
> | FAPO-32B     | **42.4** | **33.5** | **91.6** | **74.6** | **33.6 (+5.0)**   | **53.1**     | **54.8 (+3.8)** |
>
> FAPO achieves consistent improvements across a broad range of tasks, demonstrating its effectiveness and generalizing capability.
>
> > W3: Only two model families (Qwen2.5-Math-7B and Qwen2.5-32B) are tested, both starting from pre-trained models. It would be valuable to stress-test the proposed GenRM on models that already have some code-starts with long-CoT trajectories
>
> We appreciate your suggestion and have added an effectiveness evaluation on Qwen3-4B (long-cot model), as shown below:
>
> | Model           | AIME24   | AIME25   | GPQA-Diamond |
> | --------------- | -------- | -------- | ------------ |
> | Qwen3-4B + GRPO | 73.9     | 66.1     | 65.9         |
> | Qwen3-4B + FAPO | **75.4** | **66.7** | **66.2**     |
>
> In fact, we have conducted experiments on several long-CoT models. However, these models turned out to be unsuitable for informative comparison for the following reasons:
>
> - **These state-of-the-art models are difficult to further improve with existing open-source data.** Continued GRPO training can even lead to performance degradation, making method-level comparisons on such models less meaningful.
> - **Training costs are extremely high, with unstable and slow returns.** Training a 4B long-CoT model would require **more than 2×** the compute used for FAPO-32B, while (1) intermediate metrics exhibit large variance and (2) the performance improves only marginally and very slowly.
>
> We choose to start from pre-trained models because they allow us to capture the full learning dynamics throughout the entire post-training trajectory. In this context, the Instruct models can be viewed as intermediate points along this trajectory. Therefore, demonstrating that FAPO is effective on pre-trained models also suggests that it can transfer to existing post-trained models to some degree.

---

> ### Author Response · Authors · 2025-11-22
> **Rebuttal by Authors [2/N]**
>
> > W4: While the improvements are consistent, the absolute performance gains come at considerable cost: (1) requires training an additional 4B reward model, (2) needs complex asynchronous infrastructure
>
> You are correct that introducing a reward model inevitably incurs additional cost, but its role is indispensable, as:
> 1. the amount of verifiable supervision is extremely limited, and our experiments show that additional supervisory signals are necessary;
> 2. in non-verifiable settings, a reward model is indispensable.
>
> That said, **the overhead introduced by FAPO is relatively modest**, and we provide our time distribution of different RL components below:
>
>
> |                     | **Nodes** | Rollout (Inference) | **GenRM (Inference)** | Policy Update (Training) |
> | ------------------- | --------- | ------------------- | --------------------- | ------------------------ |
> | FAPO-7B             | **4**     | 42%                 | **18%**               | 33%                      |
> | FAPO-32B            | **8**     | 60%                 | **14%**               | 20%                      |
> | Qwen3-4B (Long-CoT) | **16**    | 72%                 | **10%**               | 14%                      |
>
> As the model scales and more computuing resources involved, we observe that the relative cost introduced by GenRM decreases, thanks to two design advantages:
> 1. a 4B FAPO-GenRM is fast at inference while still providing high-quality supervision to a 32B policy model
> 2. our RL infrastructure introduces the Reward Loop, which improves inference efficiency by ~30% for reward-model-based RL training.
>
> We would also like to clarify two potential misunderstandings:
> - **The complex asynchronous infrastructure is not a necessity for FAPO:** FAPO's algorithmic design (i.e., flaw-aware learning) and its infrastructure design (i.e., the asynchronous Reward Loop) are **orthogonal and independent**. The algorithm improves training effectiveness, while the infrastructure improves efficiency.
> - **Asynchronous RL infrastructure is required for large-scale RL training in general:** Async infra has been almost a necessity for modern large-scale (e.g., > 32 nodes, params > 100B) RL training and is already adopted by many leading LLM systems [1,2,3,4]. Increased infrastructure complexity (to guarantee efficiency) is therefore unavoidable. However, the open-source community currently lacks practical async RL implementations. In this context, we introduce the Reward Loop, a partially asynchronous infra design for reward models that provides a 30% speedup and is fully open-sourced.
>
> In summary, FAPO makes two independent contributions:
>
> - Algorithmic effectiveness: revealing the flawed-reasoning phenomenon and providing an effective solution with encouraging results;
> - Infrastructure scalability: the open-sourced Reward Loop enables efficient and scalable RL training in practice.
>
> Together, these contributions advance the direction of effective and scalable RL for LLMs (particularly in reward-model–based settings.).
>
> [1] Kimi k1.5: Scaling Reinforcement Learning with LLMs
>
> [2] Seed1.5-Thinking: Advancing Superb Reasoning Models with Reinforcement Learning
>
> [3] The Art of Scaling Reinforcement Learning Compute for LLMs
>
> [4] Magistral

---

> > ### Author Response · Authors · 2025-11-22
> > **Rebuttal by Authors [3/N]**
> >
> > > Q1: Why does the student model outperform the teacher model? Table 2 and Figure 3 show that FAPO-GenRM-4B outperforms Qwen3-32B (teacher) on FlawedPositiveBench
> >
> > Good Question! There are two main reasons:
> > - **Controlled randomness by consensus filtering.** During data synthesis, we sample each instance three times. A sample is kept only if all three generations yield consistent outcomes (i.e., the process errors occur at exactly the same positions). As a result, the retained samples tend to be highly reliable and have strong internal agreement. The teacher only generates once during the evaluation process.
> > - **Mission-focused training.** The distilled student model is optimized for a very specific role, i.e., the critic. This single-task specialization allows the model to concentrate its capacity on one objective, which we believe leads to stronger and more focused performance.
> > We will fully open-source the dataset, training code, and trained models to **ensure complete reproducibility**.
> >
> > > Q2: What is the overall computational cost overhead? The paper mentions training time increases by "less than 20%" but lacks comprehensive cost analysis (e.g., the cost of data synthesis, training GenRM, and detailed inference costs during RL). In particular, I am curious whether the inference cost would increase for GenRM if performing FAPO on models that have some code-starts with long-CoT
> >
> > This is an important question that we previously overlooked. Below we provide a detailed breakdown of the computational overhead,
> >
> > |                     | Rollout | Rollout Response Len | **GenRM** | GenRM Response Len | Policy Update | **Nodes** |
> > | ------------------- | ------- | -------------------- | --------- | ------------------ | ------------- | --------- |
> > | FAPO-32B            | 60%     | ~2.3k                | **14%**   | ~3.2k              | 20%           | **8**     |
> > | Qwen3-4B (Long-CoT) | 72%     | ~12k                 | **10%**   | ~3.8k              | 14%           | **16**    |
> >
> > The relative inference cost for long-cot models actually decreases, specifically:
> >
> > - Rollout: Long-CoT models exhibit a strong long-tail issue: the generation time is bounded by the longest trajectory in the batch. Therefore, rollout time increases significantly as the model generates longer traces (e.g., 12k tokens for Qwen3-4B).
> > - GenRM: In contrast, GenRM inference time does not increase with longer trajectories. We observe that the GenRM output length stays nearly constant across models (e.g., 3.2k → 3.8k).
> > - Policy update: This stage accounts for a relatively small portion of the total cost and primarily scales with model size rather than trajectory length.
> >
> > Therefore, the root cause is that the critic task for GenRM requires a nearly fixed response length (and inference time) across different settings.
> >
> > ---
> >
> > Thank you for your valuable feedback, which has greatly helped us improve our work. These new experiments and clarifications will be incorporated into our revised version soon.
> >
> > We hope that our explanations have clarified the positioning of our contributions and highlighted the exploratory efforts we have made in advancing generative reward modeling, both algorithmically and infrastructurally.
> > We look forward to further discussions with you.

---

### Official Review · Reviewer_hAMR · 2025-11-01

**Soundness:** 3
**Presentation:** 4
**Contribution:** 3
**Rating:** 6
**Confidence:** 4

**Summary:**

The paper begins by identifying a common failure mode in outcome-based RLVR: flawed-positive rollouts, which are trajectories that produce correct final answers through flawed reasoning. It first quantifies their prevalence and twofold effects during training, then proposes FAPO (Flawed-Aware Policy Optimization) as an enhancement to DAPO/GRPO based on this insight. FAPO trains a generative reward model (GenRM) with both outcome- and process-level supervision to detect flawed positives, and integrates this signal into the policy optimization process via an additional penalization term to the outcome reward. Experimentally, the trained FAPO-GenRM achieves higher F1 scores than baselines on ProcessBench and the curated FlawedPositiveBench. When applied to train Qwen-7B and Qwen-32B reasoning models, FAPO delivers improved performance on AIME24/25 and GPQA-Diamond, reducing flawed-positive ratios compared to GRPO baselines without increasing the inference token budget. Additional ablations analyze GenRM effectiveness, examine potential reward-hacking risks, and show that asynchronous deployment keeps computational overhead modest.

**Strengths:**

- **Well-motivated.** The paper clearly identifies *flawed-positive rollouts* as a pervasive yet unresolved failure mode in RLVR, illustrating their dynamics through quantitative trends (Figure 2) and motivating the need for process-level awareness beyond outcome-based rewards. This diagnosis provides a solid conceptual foundation for introducing FAPO.
- **Sound theoretical formulation.** The theoretical sections (Sec. 3.2, Appendix A) formalize the reward-penalization mechanism using group-relative advantage estimation. The analysis further justifies adopting a fixed penalty coefficient (λ = 1) based on the majority-guided condition, making FAPO both principled and free of hyperparameters.
- **Comprehensive experimental validation.** The experiments cover multiple reward benchmarks (FlawedPositiveBench, ProcessBench) and reasoning benchmarks (AIME24/25, GPQA-Diamond). The study includes one 4B model for the reward model and two models (7B and 32B) for the reasoner model training, supported by detailed analysis on FAPO-GenRM's detection ability, training effectiveness, and the reward-hacking risks. Together, these components form a coherent and comprehensive empirical evaluation.

**Weaknesses:**

- **[Soundness]** While the observations in Section 3 are insightful, the claimed causal relationship between flawed positives and performance gains (lines 190–192) requires further evidence. The presented figure only demonstrates a correlation, not causation, and additional controlled ablations would be needed to validate this conclusion.
- **[Soundness]** The evaluation datasets are relatively small. AIME (30 samples) can be insufficient for robust reasoning assessment, and on the larger GPQA-Diamond (198 samples), the observed improvement is modest (Figure 4, 1.5 points). Moreover, in Figure 4 (bottom right), the flawed-positive ratio remains high even after FAPO training, suggesting potential limits in its corrective capability.
- **[Significance]** Although the theoretical formulation is novel and well-structured, the method essentially extends RLVR by integrating outcome and process rewards into a unified training signal. Similar ideas have been explored in prior works such as [1] and [2], which slightly reduces the originality of the contribution. Furthermore, the algorithm focuses solely on mitigating flawed positives, leaving other issues in RLVR, such as false negatives, unaddressed.

[1] Process Reinforcement Through Implicit Rewards, Arxiv Feb 2025

[2] Rubrics as Rewards: Reinforcement Learning Beyond Verifiable Domains, Arxiv July 2025

**Questions:**

1. How sensitive is FAPO to the choice of λ beyond the majority-guided setting (λ = 1)? Could a curriculum that gradually increases λ during training yield better performance than the fixed choice?
2. Do the authors have the performance results on other benchmarks (i.e., AIME25, GQPA) corresponding to the setup shown in Figure 5?
3. During RL training, the GenRM remains fixed while the policy model continuously updates, which may lead to reward-hacking behavior. In Figure 7, FAPO-GenRM appears more robust than PRM, but to what extent? Has its robustness been quantitatively evaluated, and could it eventually suffer from reward hacking as well?

---

> ### Author Response · Authors · 2025-11-22
> **Rebuttal by Authors [1/N]**
>
> Thank you for your recognition of our work and for the professional suggestions.
>
> ---
>
> > W1: [Soundness] While the observations in Section 3 are insightful, the claimed causal relationship between flawed positives and performance gains (lines 190–192) requires further evidence. The presented figure only demonstrates a correlation, not causation, and additional controlled ablations would be needed to validate this conclusion.
>
> This is a cruial point. Below, we provide a more complete explanation of *how mitigating flawed positives leads to performance gains*, supported by empirical evidence and theoretical analysis.
>
> **1. Early Stage: Flawed behavior → More correct rollouts → Early performance gains**
>
> - **Flawed behavior → More correct rollouts:** Prior studies [1,2] have shown that flawed reasoning often acts as a shortcut to the correct final answer. This bias, inherited from pre-training [3], leads to certain flawed rollouts. Thus, in the early phase of RL, flawed positives naturally increase the number of correct final answer rollouts.
> - **More Correct Rollouts → early performance gains:** A larger pool of correct rollouts yields more positive rewards, providing stronger supervision and driving exploitation early in training. This effect can be directly observed in train-time reward statistics:
>
> (here we count flawed rollouts reward as 1 as it's usable in early stages)
>
> | Training Step                         | 10       | 20       | 30       | 40       | 50       |
> | ------------------------------------- | -------- | -------- | -------- | -------- | -------- |
> | Train-time reward w/o flawed rollouts | 0.14     | 0.27     | 0.29     | 0.34     | 0.36     |
> | Train-time reward w/ flawed rollouts  | **0.18** | **0.33** | **0.37** | **0.42** | **0.44** |
>
> This brings quick performance gains in the early optimization stage.
>
> **2. Later Stage: Penalizing flawed behavior → Fewer flawed rollouts → Performance gains**
>
> - **Penalizing flawed behavior → Fewer flawed rollouts:** Our theoretical analysis demonstrates the optimization distribution shift: flawed rollouts receive negative advantage, progressively reducing the model's tendency to produce flawed processes. This trend is reflected in Figure 1 (left), where the proportion of flawed positives decreases steadily as training progresses.
> - **Fewer flawed rollouts → Performance gains:** As flawed rollouts diminish, the model allocates more rollout chances to fully correct trajectories. Consequently, the RL loop receives a higher density of genuinely useful reward signals, improving final performance.
>
> (here we count flawed rollouts reward as 0 as it will reinforce flawed patterns in later stages)
>
> | Training Step                | 120      | 140      | 160      | 180      | 200      |
> | ---------------------------- | -------- | -------- | -------- | -------- | -------- |
> | Train-time reward (baseline) | 0.35     | 0.37     | 0.36     | 0.40     | 0.39     |
> | Train-time reward (FAPO)     | **0.38** | **0.40** | **0.41** | **0.44** | **0.45** |
>
> [1] Examining false positives under inference scaling for mathematical reasoning
>
> [2] Processbench: Identifying process errors in mathematical reasoning
>
> [3] why language models hallucinate

---

> ### Author Response · Authors · 2025-11-22
> **Rebuttal by Authors [2/N]**
>
> > W2: [Soundness] The evaluation datasets are relatively small. AIME (30 samples) can be insufficient for robust reasoning assessment, and on the larger GPQA-Diamond (198 samples), the observed improvement is modest (Figure 4, 1.5 points). Moreover, in Figure 4 (bottom right), the flawed-positive ratio remains high even after FAPO training, suggesting potential limits in its corrective capability.
>
> Thanks for your advice. We add more evaluation benchmarks as follows (4 math, 1 code and 1 general domain):
>
> |              | AIME24   | AIME25   | AMC      | MATH     | LiveCodeBench   | GPQA-Diamond | Avg             |
> | ------------ | -------- | -------- | -------- | -------- | --------------- | ------------ | --------------- |
> | Baseline-32B | 38.9     | 29.5     | 85.0     | 72.8     | 28.6            | 51.0         | 51.0            |
> | FAPO-32B     | **42.4** | **33.5** | **91.6** | **74.6** | **33.6 (+5.0)** | **53.1**     | **54.8 (+3.8)** |
>
> The results show that we achieve consistent improvements across all subsets, demonstrating the effectiveness.
>
> The flawed-reasoning behavior has been obviously reduced, as reflected in the improvements on AIME24 (15.5 → 7.1) and AIME25 (10.9 → 1.7). The higher error ratio on GPQA-Diamond is largely due to the nature of the task that it is a multiple-choice benchmark, so even a model with no understanding still has a 25% chance (i.e., one choice from "A,B,C,D") of guessing the correct answer.
>
> We examine the flawed-positive samples and find that many of them are essentially near-random guesses without any correct reasoning. So we tend to view these cases as not genuinely belonging to the “flawed-positive” category, as the underlying reasoning is entirely incorrect.
>
> Overall, this pattern appears to be more closely related to the model’s inherent capability limits rather than a shortcoming of FAPO’s correction mechanism.
>
> ---
>
> > W3: [Significance] Although the theoretical formulation is novel and well-structured, the method essentially extends RLVR by integrating outcome and process rewards into a unified training signal. Similar ideas have been explored in prior works such as [1] and [2], which slightly reduces the originality of the contribution. Furthermore, the algorithm focuses solely on mitigating flawed positives, leaving other issues in RLVR, such as false negatives, unaddressed.
>
> We agree that process supervision has been a promising direction, and prior works have reported encouraging results. That said, to our knowledge, FAPO is the first to **bridge and open-source the full pipeline** for training and deploying generative reward models (GRMs) effectively within RL. Our contribution can be summarized as follows:
>
> - **Algorithmic Effectiveness:** FAPO effectiveness has been verified both empirically and theoretically. Our results show that GenRM can be robust to reward hacking and bring substantial improvements. [1] is representative work on discriminative reward models, and our results demonstrate that GenRM exhibits stronger robustness, as shown in the experiments discussed in our responses to Q2.
> - **Infrastructure Efficiency:** We design an efficient RL reward-loop infrastructure that significantly accelerates reward computation. In the reward-model setting, this leads to approximately 30% speedup. While [2] also employs a form of generative rewards, its associated resources are not open-sourced, making it difficult to reproduce or extend.
> - **Open-Source Recipe:** FAPO fully open-sources the data, models, and usable RL infrastructure, providing an end-to-end reproducible recipe. We believe this offers a valuable reference implementation that can serve as an important baseline for future research.
>
> [1] Process Reinforcement Through Implicit Rewards
>
> [2] Rubrics as Rewards: Reinforcement Learning Beyond Verifiable Domains

---

> ### Author Response · Authors · 2025-11-22
> **Rebuttal by Authors [3/N]**
>
> > Q1: How sensitive is FAPO to the choice of λ beyond the majority-guided setting (λ = 1)? Could a curriculum that gradually increases λ during training yield better performance than the fixed choice?
>
> Good Question. We experiment with more $\lambda$, and list the results as below.
>
> | Setting                                                        | Performance     |
> | -------------------------------------------------------------- | --------------- |
> | Baseline-7B ($\rho = +\infty \Rightarrow \lambda = 0$)         | 32.1            |
> | FAPO with $\rho = 2 \Rightarrow \lambda = 1/3$                 | 34.6            |
> | FAPO with $\rho = 1 \Rightarrow \lambda = 1$ (default setting) | 36.8 (reported) |
> | FAPO with $\rho = 1/2 \Rightarrow \lambda = -1/3$              | 39.6            |
>
> We can conclude that:
> - flaw-aware learning consistently improves performance, as all configurations outperform the baseline.
> - Achieving the best performance require tuning the parameter $\lambda$. In the 7B setting, a more aggressive strategy leads to larger gains. The configuration with $\rho = 1/2$, which corresponds to an optimization shift where about one third of the rollouts are fully correct, achieves the best performance.
>
> So overall, flaw-aware learning leads to performance gains, but achieving the best performance requires tuning the parameter $\lambda$. That said, FAPO introduces only this single additional parameter, which makes the tuning process relatively easy.
>
> ---
>
> > Q2: Do the authors have the performance results on other benchmarks (i.e., AIME25, GQPA) corresponding to the setup shown in Figure 5?
>
> We suppliment the results as follows:
>
> | Model                        | AIME24 | AIME25 | GPQA |
> | ---------------------------- | ------ | ------ | ---- |
> | Baseline-7B                  | 32.1   | 17.7   | 31.8 |
> | FAPO-7B (w/ Baseline GenORM) | 34.9   | 18.5   | 32.0 |
> | FAPO-7B (w/ FAPO-GenRM)      | 36.8   | 18.1   | 32.8 |
>
> ---
>
> > Q3: During RL training, the GenRM remains fixed while the policy model continuously updates, which may lead to reward-hacking behavior. In Figure 7, FAPO-GenRM appears more robust than PRM, but to what extent? Has its robustness been quantitatively evaluated, and could it eventually suffer from reward hacking as well?
>
> Excellent question. We attempted to quantify the robustness gap between discriminative PRM (DisPRM) and FAPO-GenRM by training them on the same data and then applying both models in RL training. Our observations are as follows:
>
> - First of all, **the training of GenPRM is more robust than DisPRM**. Because the training data inevitably contains some noise, we found that DisPRM tends to overfit more easily, leading to degraded performance on the held-out evaluation.
>
> |             | Best Checkpoint | Last Checkpoint |
> | ----------- | --------------- | --------------- |
> | DisRM       | 82.4            | 74.3 (overfit)  |
> | FAPO-GenPRM | 90.5            | 90.0            |
>
> - When integrated into the final RL training, we consistently observe: **FAPO-GenRM > Baseline > PRM**. This further confirms that PRM does not always provide benefits, whereas the generative reward signal offers substantially stronger gains.
>
> |                   | AIME24 | AIME25 | GPQA-Diamond |
> | ----------------- | ------ | ------ | ------------ |
> | Baseline-32B      | 38.9   | 29.5   | 51.0         |
> | FAPO-32B + DisPRM | 36.8   | 26.1   | 48.9         |
> | FAPO-32B + GenPRM | 42.4   | 33.5   | 53.1         |
>
> Based on the above experiments, we can conclude that (1) **FAPO-GenRM trained on the same data is consistently stronger than DisPRM**, which is also aligned with findings in several recent works [1,2], and (2) **FAPO-GenRM provides a higher-quality reward signal for RL**, and across all our experiments, we have not observed any reward-hacking behavior.
>
> [1] GenPRM: Scaling Test-Time Compute of Process Reward Models via Generative Reasoning
>
> [2] Inference-Time Scaling for Generalist Reward Modeling
>
> [3] Process Reward Models That Think
>
> ---
>
> Thanks for your great suggestions, which have been very helpful in improving our work.
> We hope that our responses address your concerns. If you have any further questions about our paper or our rebuttal responses, please do not hesitate to reach out; we would be very happy to clarify them for you.

---

### Official Review · Reviewer_wvzj · 2025-11-01

**Soundness:** 3
**Presentation:** 3
**Contribution:** 3
**Rating:** 6
**Confidence:** 2

**Summary:**

This paper addresses the issue of "flawed-positive" rollouts in reinforcement learning with verifiable rewards (RLVR) for LLMs, where models arrive at correct answers via unreliable reasoning (e.g., guessing). The authors first analyze this phenomenon, finding that while flawed positives provide rapid early gains, they ultimately constrain capability by reinforcing these unreliable patterns. To mitigate this, they propose Flawed-Aware Policy Optimization (FAPO), a method that applies a parameter-free reward penalty to flawed-positive rollouts. This approach aims to leverage flawed positives as shortcuts during the initial warm-up phase while gradually shifting the optimization objective toward reliable reasoning in the later refinement stage. To enable this, the work also introduces a generative reward model (GenRM) trained with a process-level reward to accurately detect and localize reasoning errors within rollouts. The authors claim FAPO improves correctness, reliability, and stability without increasing the token budget.

**Strengths:**

1. The paper clearly identifies and analyzes a critical problem in RLVR. The preliminary study (Section 2.2) effectively demonstrates the "twofold effect" of flawed positives—acting as "stepping stones" early in training but hindering optimization later —providing a solid empirical foundation for the proposed solution.

2. The FAPO algorithm's adaptive reward penalty is simple yet theoretically grounded. The analysis in Appendix A demonstrates how this mechanism creates an automatic, parameter-free "optimization shift".

3. The development of a compact (4B) generative reward model (GenRM) is a key strength. The step-wise, distance-sensitive reward formulation used to train it is well-designed to encourage precise error localization rather than simple binary guessing. This trained GenRM is shown to be highly effective, outperforming its 32B teacher model and a 72B SOTA discriminative model.

4. The experiments are thorough, testing on multiple models (7B, 32B) and benchmarks (AIME24, AIME25, GPQA). Crucially, the authors present full learning curves rather than just final checkpoints. This transparently supports claims of improved training stability , outcome correctness , and process reliability (a reduced flawed positive ratio).

**Weaknesses:**

1. The entire framework's effectiveness is contingent on the quality of the FAPO-Critic-85K dataset, which was labeled by a "teacher model" (Qwen3-32B). The GenRM can only learn to detect flaws that the teacher model can identify. This creates a fundamental performance ceiling; any subtle errors missed by the teacher will be propagated, and FAPO will fail to penalize them.

2. The GenRM is trained primarily on mathematical reasoning tasks. While it shows good performance on GPQA-Diamond, it is unclear how well this math-trained critic can generalize to detecting a wider array of non-mathematical flawed reasoning (e.g., logical fallacies, factual inconsistencies) in open-domain tasks. The paper acknowledges this as a limitation.

3. The proposed asynchronous architecture, while necessary for performance, adds significant infrastructure complexity compared to standard RLVR, which uses a simple, synchronous rule-based verifier. The authors note this adds a non-trivial training time increase of "less than 20%", which could be a barrier to adoption.

**Questions:**

none

---

> ### Author Response · Authors · 2025-11-22
> **Rebuttal by Authors [1/N]**
>
> Thank you so much for your review and insightful, constructive feedback. Your suggestions have been invaluable for improving the paper's clarity and rigor.
>
> ---
>
> > W1: The entire framework's effectiveness is contingent on the quality of the FAPO-Critic-85K dataset, which was labeled by a "teacher model" (Qwen3-32B). The GenRM can only learn to detect flaws that the teacher model can identify. This creates a fundamental performance ceiling; any subtle errors missed by the teacher will be propagated, and FAPO will fail to penalize them.
>
> Yes, you have pointed out a key issue in knowledge distillation that the student model's performance is inherently bounded by the teacher model. While this limitation is unavoidable, FAPO incorporates several strategies to mitigate its impact:
>
> - **Consensus filtering:** During data synthesis, we sample each instance three times. A sample is kept only if all three generations yield consistent outcomes (i.e., the process errors occur at exactly the same step or no errors at all). As a result, the retained samples tend to be highly reliable and have strong internal agreement.
> - **Robust Training Objective:** Our designed training objective (in Equation 7) are robust to the subtle errors missed by the teacher. The reward supervision signal is a soft noise-robust label, the student can still receive an appropriate reward even when the annotated error location deviates slightly from the true error.
>
> We will include the relevant details and discussions in the revised version.
>
> > W2: The GenRM is trained primarily on mathematical reasoning tasks. While it shows good performance on GPQA-Diamond, it is unclear how well this math-trained critic can generalize to detecting a wider array of non-mathematical flawed reasoning (e.g., logical fallacies, factual inconsistencies) in open-domain tasks. The paper acknowledges this as a limitation.
>
> Thank you for the insightful comment. **Generalization** is one of the key advantages of RL compared to standard supervised finetuning, **in both FAPO-Reasoning and FAPO-GenRM**.
>
> We extend **FAPO-Reasoning** to 2 more math benchmarks and 1 code benchmarks, with the overall results as below:
>
> |              | AIME24   | AIME25   | AMC      | MATH     | LiveCodeBench   | GPQA-Diamond | Avg             |
> | ------------ | -------- | -------- | -------- | -------- | --------------- | ------------ | --------------- |
> | Baseline-32B | 38.9     | 29.5     | 85.0     | 72.8     | 28.6            | 51.0         | 51.0            |
> | FAPO-32B     | **42.4** | **33.5** | **91.6** | **74.6** | **33.6 (+5.0)** | **53.1**     | **54.8 (+3.8)** |
>
> The results show that we achieve consistent improvements across all subsets and FAPO-Reasoning can generalize to other tasks effectively.
>
> For **FAPO-GenRM in open-ended tasks**, obtaining large-scale datasets with reliable evaluation protocols is challenging. To provide an assessment, we manually sampled 40 examples from RewardBench (covering a diverse set of tasks and responses). We then used GenORM and FAPO-GenRM to evaluate these examples respectively and conducted an A/B test in which human annotators judged which model’s assessment was preferred, with an absolute score for FAPO-GenRM ranging from 0 to 10.
>
> A/B Test: FAPO-GenRM vs. GenORM
>
> | Win | Tie  | Loss |
> | --- | ---- | ---- |
> | 70  | 17.5 | 12.5 |
>
> Absolute mean score of FAPO-GenRM = 7.6 / 10.
>
> **So overall, both FAPO-GenRM and FAPO-Reasoning can generalize to other tasks effectively.**

---

> ### Author Response · Authors · 2025-11-22
> **Rebuttal by Authors [2/N]**
>
> > W3: The proposed asynchronous architecture, while necessary for performance, adds significant infrastructure complexity compared to standard RLVR, which uses a simple, synchronous rule-based verifier. The authors note this adds a non-trivial training time increase of "less than 20%", which could be a barrier to adoption.
>
> We would like to correct a potential misunderstanding. FAPO's algorithmic design (i.e., flaw-aware learning) and its infrastructure design (i.e., the asynchronous reward loop) are orthogonal and independent. The algorithmic component improves training effectiveness, while the infrastructure component improves training efficiency.
>
> The asynchronous reward loop design in FAPO improves training efficiency *without altering any intermediate computational results.*
>
> To further clarify FAPO's scaling potential in large-scale RL systems, we future offer the following clarifications:
>
> 1. **FAPO application in large-scale RL systems**
>
> In larger-scale settings, the additional burden introduced by FAPO GenRM remains relatively small. Below we provide a more detailed breakdown of the time distribution across different RL stages:
>
> |                     | **Nodes** | Rollout (Inference) | **GenRM (Inference)** | Policy Update (Training) |
> | ------------------- | --------- | ------------------- | --------------------- | ------------------------ |
> | FAPO-7B             | **4**     | 42%                 | **18%**               | 33%                      |
> | FAPO-32B            | **8**     | 60%                 | **14%**               | 20%                      |
> | Qwen3-4B (Long-CoT) | **16**    | 72%                 | **10%**               | 14%                      |
>
> As the model size increases and more computing resources are involved, the relative overhead of GenRM continues to decrease. Therefore, in larger systems, the cost of FAPO GenRM becomes even smaller.
>
> 2. **Infrastructure Complexity of large-scale RL systems**
>
> Asynchronous RL infrastructure has become a necessity for modern large-scale RL training and is already adopted by many leading LLM systems [1,2,3] for better efficiency. Increased infrastructure complexity is therefore unavoidable.
>
> However, the community currently lacks practical async RL implementations. In this context, we introduce the Reward Loop, a partially asynchronous infra design for reward models that provides a 30% speedup and is fully open-sourced.
>
> [1] Kimi k1.5: Scaling Reinforcement Learning with LLMs
>
> [2] Seed1.5-Thinking: Advancing Superb Reasoning Models with Reinforcement Learning
>
> [3] The Art of Scaling Reinforcement Learning Compute for LLMs
>
> [4] Magistral

---

> > ### Comment · Reviewer_wvzj · 2025-11-24
> >
> > Thank you for your reply. I believe my questions have been resolved, so I have decided to increase my rating.

---

> > > ### Author Response · Authors · 2025-11-27
> > > **Author Response**
> > >
> > > We appreciate your feedback and your positive assessment of our work. We have incorporated these experimental results and the corresponding discussion into the revised version.

---

### Official Review · Reviewer_DJdo · 2025-11-01

**Soundness:** 3
**Presentation:** 2
**Contribution:** 3
**Rating:** 4
**Confidence:** 3

**Summary:**

This paper addresses a critical issue in RLVR for LLMs: the reinforcement of flawed reasoning patterns. In RLVR, models are rewarded for generating trajectories that lead to a correct final answer. However, many such positive rollouts contain unreliable reasoning steps, which are rewarded identically to fully correct solutions. This flawed-positive problem can lead to models that are correct on a specific metric but ultimately unreliable.

The authors first conduct a systematic analysis, revealing that flawed positives act as valuable stepping stones for rapid early learning but later constrain performance by reinforcing bad habits. To address this, they propose FAPO, a two-stage solution: They train a efficient generative reward model using a novel step-wise RL objective that localizes the first error in a reasoning chain. This model achieves state-of-the-art performance on error detection benchmarks. FAPO applies a parameter-free reward penalty to rollouts flagged as flawed-positive by the GenRM. The penalty is designed to dynamically shift the learning focus, initially allowing flawed positives to aid learning before gradually steering the policy toward fully reliable reasoning as its capability improves.

**Strengths:**

The problem tackled is of great significance to the LLM reasoning community. As RLVR becomes a dominant paradigm for advancing LLM capabilities, ensuring that the learned reasoning is not just correct but also reliable and transparent is crucial for safety and trustworthiness. FAPO provides a practical, efficient, and theoretically grounded solution that improves both the efficiency and the final quality of RL training. The release of code and benchmarks further enhances its impact. Experiments on mathematical and GPQA reasoning tasks demonstrate that FAPO improves outcome correctness, reduces the rate of flawed positives, and enhances training stability without increasing response length.

**Weaknesses:**

While the results on mathematical reasoning and GPQA are strong, the paper's claims of broad domains would be more convincing with validation on a wider range of tasks. A key domain of interest is code generation, where verifiable rewards are common and flawed reasoning (e.g., code that passes specific tests but is buggy or inefficient) is a major concern. Demonstrating coding effectiveness would significantly broaden the method's impact and generalizability.

The 20% training-time overhead, while reasonable, could be a barrier to truly massive-scale training. A more detailed discussion of the bottlenecks and potential optimizations for a synchronous system (even as future work) would be valuable for practitioners looking to adopt it at a larger scale.

The paper focuses on tasks with easily verifiable final answers. A natural question is how FAPO would be adapted to more subjective or open-ended tasks where a "correct" final answer is not binarily verifiable. In such scenarios, would the process-level reward become the primary signal?

**Questions:**

Please refer to "Weaknesses".

---

> ### Author Response · Authors · 2025-11-22
> **Rebuttal by Authors [1/N]**
>
> Thanks for your constructive suggestions and the opportunity to address your concerns.
>
> ---
>
> > W1: While the results on mathematical reasoning and GPQA are strong, the paper's claims of broad domains would be more convincing with validation on a wider range of tasks. A key domain of interest is code generation, where verifiable rewards are common and flawed reasoning (e.g., code that passes specific tests but is buggy or inefficient) is a major concern. Demonstrating coding effectiveness would significantly broaden the method's impact and generalizability.
>
> This is a crucial point. We have extend to one more code reasoning tasks (livecodebench) and two more math tasks (amc and math) as below:
>
> |              | AIME24   | AIME25   | AMC      | MATH     | **LiveCodeBench** | GPQA-Diamond | Avg             |
> | ------------ | -------- | -------- | -------- | -------- | ----------------- | ------------ | --------------- |
> | Baseline-32B | 38.9     | 29.5     | 85.0     | 72.8     | 28.6          | 51.0         | 51.0            |
> | FAPO-32B     | **42.4** | **33.5** | **91.6** | **74.6** | **33.6 (+5.0)**   | **53.1**     | **54.8 (+3.8)** |
>
> FAPO outperforms baseline in code reasoning tasks by a large margin (+5.0) and achieves consistent improvements across a broad range of tasks.
>
> ---
>
> > W2: The 20% training-time overhead, while reasonable, could be a barrier to truly massive-scale training. A more detailed discussion of the bottlenecks and potential optimizations for a synchronous system (even as future work) would be valuable for practitioners looking to adopt it at a larger scale.
>
> Thank you for pointing this out. We offer the following two clarifications:
>
> 1. **FAPO application in large-scale RL systems:**
>
> In larger-scale RL training settings, the additional burden introduced by FAPO GenRM remains relatively small. Below we provide a more detailed breakdown of the time distribution across different RL stages:
>
> |                     | **Nodes** | Rollout (Inference) | **GenRM (Inference)** | Policy Update (Training) |
> | ------------------- | --------- | ------------------- | --------------------- | ------------------------ |
> | FAPO-7B             | **4**     | 42%                 | **18%**               | 33%                      |
> | FAPO-32B            | **8**     | 60%                 | **14%**               | 20%                      |
> | Qwen3-4B (Long-CoT) | **16**    | 72%                 | **10%**               | 14%                      |
>
> As the model size increases and rollout trajectories become longer, the relative overhead of GenRM continues to **decrease**, because
> - The cost introduced by FAPO GenRM is relatively fixed,
> - Our Reward Loop infrastructure design enables efficient scaling.
>
> 2. **Bottlenecks and possible optimizations for synchronous systems:**
>
> As we illustrate in Figure 8, the long-tail problem has long been a key bottleneck in scaling large-scale RL systems where GPUs often remain idle during the generation of long-tail samples, hindering the scaling of RL training.
>
> There are two main directions for improvement at larger scale:
>
> - One promising (but challenging) direction is to predict potential long-tail trajectories and proactively utilize idle GPUs to process them. This requires substantial infrastructure complexity and careful system design.
> - Another widely used approach is fully-async training [1-5], where each role runs in standalone GPU resources. In this context, we introduce the **partially asynchronous reward loop**, which reduces reward-modeling time by **approximately 30%**, and we have **open-sourced** the entire implementation.
>
> We consider fully-async RL framework as a promising direction and will continue building fully async infrastructure with effective algorithms in the future.
>
> [1] Kimi k1.5: Scaling Reinforcement Learning with LLMs
>
> [2] Seed1.5-Thinking: Advancing Superb Reasoning Models with Reinforcement Learning
>
> [3] The Art of Scaling Reinforcement Learning Compute for LLMs
>
> [4] Magistral
>
> [5] AReaL: A Large-Scale Asynchronous Reinforcement Learning System for Language Reasoning

---

> > ### Author Response · Authors · 2025-11-22
> > **Rebuttal by Authors [2/N]**
> >
> > > W3: The paper focuses on tasks with easily verifiable final answers. A natural question is how FAPO would be adapted to more subjective or open-ended tasks where a "correct" final answer is not binarily verifiable. In such scenarios, would the process-level reward become the primary signal?
> >
> > We appreciate your interest in FAPO effectiveness in more subjective or open-ended tasks. These tasks has been a fundamental challenge in reinforcement learning. We believe the following components are particularly important:
> >
> > - Generative Reward: GenRMs provide interpretable feedback and is generally more robust to reward hacking in RL. Our proposed reward-loop design offers a practical infrastructure to support this mechanism.
> >
> > - Human-aligned evaluation granularity: The model needs to generate task-appropriate subjective evaluation dimensions, such as fluency or coherence. Alternatively, the model can be prompted to propose critic dimensions and critique its own outputs, as adopted in DeepSeek-GRM [1].
> >
> > Our future work will focus on training generalist reward models and exploring their applications in reinforcement learning.
> >
> > [1] Inference-Time Scaling for Generalist Reward Modeling
> >
> > ---
> >
> > Thank you for your valuable comments. We hope our response has addressed your concerns.
> >
> > We also look forward to your response and are glad to engage further on both the algorithmic and infrastructural aspects of FAPO, as well as future GenRM applications in large-scale RL systems.

---

### Author Response · Authors · 2025-12-01
**Summary of the Review and Rebuttal [Part 1]**

Dear Area Chair

Thank you for reviewing our submission. Below, we provide a concise summary of the reviews and our rebuttal process with the four asigned reviewers (with ID=DJdo, wvzj, hAMR, and Gwxh), including
(1) **positive consensus** reached across reviewers, and (2) **key concerns** raised and **our responses**.

### (1) Positive Consensus across the Reviews

**Flawed Reasoning as a Significant yet Under-explored Challenge in RL:**

Our work exposes flawed reasoning behaviors evolving in RL, and the importance of this problem is consistently acknowledged by all reviewers.

*Reviewer DJdo*:
> "The problem tackled is of great significance to the LLM reasoning community."

*Reviewer wvzj*:
> "The paper clearly identifies and analyzes a critical problem in RLVR."

*Reviewer hAMR*:
> "The paper clearly identifies flawed-positive rollouts as a pervasive yet unresolved failure mode in RLVR ..."

*Reviewer Gwxh*:
> "... provides valuable insights into the prevalence and evolution of flawed-positive rollouts ..."

**FAPO Addresses This Challenge Effectively, Both Theoretically and Empirically**

*Reviewer DJdo:*
> - Theoretically: "FAPO provides a practical, efficient, and **theoretically grounded** solution that improves both the efficiency and the final quality of RL training."
> - Empirically: "Experiments on mathematical and GPQA reasoning tasks demonstrate that FAPO improves outcome correctness, reduces the rate of flawed positives ..."

*Reviewer wvzj:*
> - Theoretically: "The FAPO algorithm's adaptive reward penalty **is simple yet theoretically grounded**."
> - Empirically: "The experiments are **thorough**, testing on multiple models (7B, 32B) ..."

*Reviewer hAMR:*
> - Theoretically: "**Sound theoretical formulation.** The theoretical sections ..."
> - Empirically: "**Comprehensive experimental validation.** The experiments cover ..."

*Reviewer Gwxh:*
> - "The effectiveness of FAPO is demonstrated both **theoretically** and **empirically**."

**Taken together**, *the reviews reflect a consensus that FAPO identifies an important yet under-explored problem in RL reasoning and provides an effective, well-validated solution to it.*

---

> ### Author Response · Authors · 2025-12-01
> **Summary of the Review and Rebuttal [Part 2]**
>
> ### (2) Summary of Reviewers' Concerns and Our Responses
>
> > Main Concern 1: FAPO's evaluation in the broader domain
>
> Reviewers expressed satisfaction with the reported results (in two math tasks and one general QA task), and showed strong interest in its potential effectiveness in more other domains (e.g., code reasoning suggested by *Reviewer DJdo*, broader math tasks proposed by *Reviewer Gwxh*, and the generalization of FAPO-GenRM highlighted by *Reviewer wvzj*.)
>
> **Summary of our response:**
>
> Following their suggestions, we have extended our evaluation as follows:
>
> - **Code Reasoning(LiveCodeBench)**, as recommended by *Reviewer DJdo*
> - **Two more Math Tasks (MATH, AMC)**, as recommended by *Reviewer Gwxh*
> - **FAPO-GenRM on open-ended critic tasks**, as recommended by *Reviewer wvzj*
>
> In the revised version:
>
> - **FAPO-GenRM** is evaluated on *ProcessBench*, *FlawedPositiveBench*, and *open-ended critic tasks*.
> - **FAPO-Reasoning** is evaluated on **four math tasks** (AIME24, AIME25, MATH, AMC), **one code task** (LiveCodeBench), and **one general-domain task** (GPQA).
>
> **Effectiveness:** Our experiments demonstrate that FAPO generalizes well to these broader domains and consistently outperforms strong baselines. During the rebuttal period, *Reviewer wvzj* responded to our rebuttal and expressed satisfaction with the results.
>
> > Main Concern 2: Additional inference-time overhead introduced by FAPO-GenRM
>
> *Reviewers DJdo, wvzj, and Gwxh* expressed concerns regarding the inference cost introduced by the reward model and the scalability of our approach. We offer the following clarifications:
>
> **Summary of our Response:**
>
> - **Inference overhead is unavoidable but reasonable and promising:**
>     - *Unavoidable:* Introducing a reward model inevitably adds training cost, which is inherent to all RM-based RL methods.
>     - *Promising*: Flawed reasoning is a critical issue for current LLMs [1], and several strong recent works have already explored reward-model–based solutions [2].
> - **Our contributions to reducing overhead and enabling scalable RL:**
>     - **Asynchronous infrastructure design:** We introduce the *Reward Loop*, which reduces reward-modeling time by approximately 30% and is fully **open-sourced**.
>     - **Scalable FAPO Experiments**: Our additional experiments show that the relative overhead of GenRM **decreases as computing resources scale (tested up to 16 nodes)**, indicating that FAPO remains practical and scalable in large-scale training scenarios.
>
> > Main Concern 3: Prior work has partially explored process rewards
>
> *Reviewers hAMR and Gwxh* pointed out several papers involving process rewards and questioned the novelty of our work in this regard.
>
> **Summary of our Response:**
>
> We agree with the reviewers that process-level supervision is a promising direction with several strong existing works.
> That said, we believe our work **differs substantially from prior studies in several key aspects**:
>
> - **Algorithmic contribution:** FAPO explores the potential of *generative process rewards*, a form of process supervision highlighted as a promising paradigm in prior discussions [2]. Most related work mentioned by the reviewers uses discriminative rewards. We show that generative rewards provide richer supervisory signals and exhibit robustness to reward hacking. We believe our method and discovery are *novel, meaningful, and promising*.
> - **Infrastructure contribution:** We introduce **Reward Loop**, an infrastructure design that substantially accelerates reward computation. Under the reward-model setting, it yields roughly a 30% speedup. *We further scale the system to 16 nodes and observe favorable scaling behavior*.
> - **Open-source contribution:** FAPO fully open-sources the data, models, and RL infrastructure, offering an end-to-end reproducible pipeline, **which is rarely achieved in previous related work**. We believe FAPO provides a valuable implementation that can serve as a strong baseline for future research on generative rewards reinforcement learning (GenRM-RL).
>
> Overall, our work advances generative process rewards from the **algorithmic, infrastructure, and open-source perspectives**. We believe these contributions collectively establish the **novelty and significance** of our approach.
>
> [1] Why Language Models Hallucinate. OpenAI.
>
> [2] DeepSeekMath-V2: Towards Self-Verifiable Mathematical Reasoning. DeepSeek AI. (*very recent work after ICLR submission*)
>
> ---
>
> All of the above revisions have been incorporated into the revised version, with the following updates:
> - Expanded experiments demonstrating FAPO’s effectiveness across broader domains
> - Detailed presentation of FAPO’s infrastructure design (Reward Loop) and scalability experiments
> - Extended ablation studies and additional discussions.
>
> Thank you again for reviewing our manuscript and for considering the discussion with the reviewers.

---

### Meta-Review · Area_Chair_K6gb · 2026-01-07

**Summary:**

The paper studies the reinforcement learning with verifiable reward. Authors identified the major issue is with the flawed-positive rollouts, and propose Flawed-Aware Policy Optimization (FAPO), which applies an adaptive, parameter-free penalty to flawed-positive rollouts detected by a compact generative reward model (GenRM) trained with process-level supervision. Reviewer raised the question of evaluation on code generation benchmarks, if the method scales with the model size, and on 20% training overhead. Another reviewer asked about the causation between found correlation. During the rebuttal, authors provided more evaluations and motivated the infrastructure cost increase. With this, the paper is close on the boarder line, however, considering that at least one reviewer increased the score to 8, my recommendation is to accept the paper.

**Reviewer Concerns:**

All reviewers asked for code related benchmarks, and authors provided strong results on Livecodebench as well as more math-related datasets. Also, multiple reviewers asked about the hit on training performance, authors provided a feedback on this by comparing with long-CoT. Some concerns might have not being answered, such as teacher-labeled data, infrastructure complexity and incremental novelty.

**Reviewer Scores:**

DJdo 4 -> 5,6 as the code benchmark was updated
wvzj 6 -> 7,8 mentioned in the discussion
hAMR 6 -> 6 the question on more evaluation benchmarks was answered.
Gwxh 4 -> 4,5

---

### Decision · Program_Chairs · 2026-01-26

Accept (Poster)